# PALADIN: PRIVACY-AWARE LEARNING WITH ADVERSARY-DETECTION AND INFERENCE SUPPRESSION

## ABSTRACT

Agents trained via Reinforcement Learning (RL) and deployed in sensitive settings, such as finance, autonomous driving, or healthcare, risk leaking private information through their observable behaviour. Even without access to raw data or model parameters, a passive adversary may infer sensitive attributes (e.g., identity, location) by observing the agent's trajectory. We empirically study this *behavioural leakage* threat and propose **PALADIN**, a proactive privacy-shaping framework that integrates an adversarial inference model into the training loop. PALADIN jointly trains a transformation network to perturb observations and a co-adaptive leakage predictor, whose output shapes the agent's reward via a curriculum-guided penalty. This allows the agent to first learn stable task policies, then progressively adapt its behaviour to resist inference. We evaluate PALADIN on an Autonomous Vehicle (AV) and a Financial Trading (FT) benchmarks. We audited leakage with held-out adversaries (MLP, GRU, and Transformer) using multiple metrics (confidence, negative log-likelihood, F1, and AUROC). On AV–GPS, PALADIN achieves strong privacy–utility improvements where in a representative MLP–Transformer setting, it increases return from 25.3 to 40.9 while reducing Attack F1 from 0.96 to 0.14 and sharply lowering adversary confidence. On the financial benchmark, gains are smaller but still positive, for example, in a GRU–Transformer setting, PALADIN increases return from 0.005 to 0.041 while slightly reducing Attack F1 from 0.75 to 0.73 and improving leak_nll. Overall, our results show that behaviour-aware, curriculum-guided shaping is highly effective for reducing behavioural leakage in AV control and offers a principled, empirically robust alternative to Differential Privacy (DP)-style methods for other sequential decision problems.

## 1 INTRODUCTION

The rapid adoption of intelligent systems in safety-critical and data-sensitive domains has intensified the challenge of balancing task performance with information privacy (Dwork et al. (2006); Abadi et al. (2016)). Domains such as autonomous driving, finance, and healthcare increasingly deploy agents that rely on sensitive data for real-time decision-making (Heaton et al. (2017); Miotto et al. (2018)). However, these agents often operate in *adversarially observable environments* (Goodfellow et al. (2014b)), where emitted behaviours, such as actions, trajectories, or timing patterns, can be exploited to infer sensitive information.

While existing approaches such as DP (Abadi et al. (2016)), homomorphic encryption (Aono et al. (2017)), and secure multi-party computation (Shokri & Shmatikov (2015)) offer strong training-time guarantees, they fail to prevent *behavioural leakage* at deployment: the unintended exposure of sensitive user information through an agent's observable actions and trajectories. This threat can arise in adversarially observable environments (Goodfellow et al. (2014b)), where a passive observer can monitor agent emissions over time. For example, an adversary observing the GPS traces of a taxi fleet may infer passenger identity; a financial bot's trade timings may reveal proprietary strategies (Hilprecht et al. (2019); Carlini et al. (2021)). Crucially, these threats operate *post hoc*, differentiating them from classical DP (which protects training data) and adversarial robustness (which defends against performance degradation via perturbation). Recent RL work has extended DP guarantees to policies (Yang-Zhao & Ng (2024); Balle et al. (2016)), injecting noise into gradients or trajectories.

However, these methods target population-level or training-time privacy, not trajectory-level semantic cues. Adversarial RL (Pinto et al. (2017); Gleave et al. (2020)) focuses on robustness to environmental shifts or reward attacks, but not inference suppression. Similarly, Information-theoretic methods (Zhao et al. (2020); Noorbakhsh et al. (2024)) limit mutual information in supervised settings, but are not optimised for online, temporally evolving agent behaviour. As a result, current methods leave agents vulnerable to deployment-time leakage.

We propose **PALADIN** (**P**rivacy-**A**ware **L**earning through **A**dversarial **D**efence and **IN**ference suppression), a proactive RL framework that suppresses behavioural leakage. PALADIN embeds a co-trained adversarial inference model into the reward loop, providing real-time estimates of semantic leakage. It shapes the policy to minimise this risk through a *curriculum-guided privacy penalty*: early training prioritises task mastery, while later phases introduce increasingly strict privacy shaping. This two-phase process is critical. Techniques such as uniform noise injection and fixed penalties destabilise learning and underperform on the privacy–utility trade-off. PALADIN instead adapts dynamically to the adversary's performance, modulating penalties based on empirical inference risk. This makes it compatible with real-world RL settings where leakage evolves during learning and no single noise level is optimal. We instantiate PALADIN on two real-world benchmarks, i) **AV-GPS dataset** (Abrar et al. (2024)): an autonomous driving task where GPS traces can leak spoofing status; and ii)**a Financial Trading dataset constructed from Yahoo Finance OHLCV** (Aroussi): a multi-asset sequential decision environment where daily price-volume trajectories can leak a latent volatility or future movement label. PALADIN consistently improves the privacy–utility trade-off, for example, on AV–GPS, PALADIN can simultaneously improve task performance and substantially reduce the success of strong held-out attackers. For example, in the MLP–Transformer pairing (Table 1), PALADIN increases return from $25.3$ to $40.9$ while reducing Attack F1 from $0.96$ to $0.14$ and sharply lowering adversary confidence. On the finance benchmark, PALADIN achieves similar, albeit smaller, improvements. In a representative GRU–Transformer configuration (Table 5), PALADIN increases return from $\approx 0.01$ to $\approx 0.04$ while slightly reducing Attack F1 from $\approx 0.75$ to $\approx 0.73$ and improving negative log-likelihood. In both domains, PALADIN consistently dominates DP-based baselines (Differentially Private-Reinforcement Learning (DP-RL) and Differentially Private with NASH Equilibrium (DP-NASH)) on the privacy–utility frontier and, in the most challenging behavioural leakage settings, outperforms static noise injection, which either leaves leakage largely unresolved or degrades utility.

For reproducibility, we will release our code upon acceptance. Hence, the contributions of this work are threefold:

- We define and empirically study *deployment-time behavioural leakage* as a distinct privacy threat in RL, which persists even after robustness or training-time DP defences.
- We propose **PALADIN**, a curriculum-guided, adversary-in-the-loop framework that proactively shapes policy behaviour to suppress semantic leakage while preserving utility.
- We provide extensive empirical validation on AV and FT benchmarks, showing consistent privacy–utility trade-offs against strong static, adversarial, and DP baselines.

## 2 RELATED WORK

Early efforts to protect sensitive information in learning systems are dominated by DP mechanisms and adversarial inference attacks. Approaches such as Differentially Private-Stochastic Gradient Descent (DP-SGD) (Abadi et al. (2016)), while providing formal guarantees, often reduce task utility and introduce considerable computational overhead (Shokri & Shmatikov (2015); Li et al. (2024)). Alternative cryptographic techniques, such as homomorphic encryption (Aono et al. (2017)) and secure multi-party computation (Shokri & Shmatikov (2015)), provide even stronger formal guarantees. However, the high computational costs and latency render them impractical for real-time applications such as autonomous driving, financial trading, or edge computing. Critically, these classical approaches focus primarily on training-time or parameter-level privacy, and fail to mitigate *behavioural leakage*, such as inference from an agent's emitted actions at deployment time.

In RL, DP has been applied to rewards (Balle et al. (2016)) and noisy gradients (Yang-Zhao & Ng (2024)), but guarantees remain at the parameter level (Sajed & Sheffet (2019); Qiao & Wang (2023)). In contrast, behavioural leakage arises when adversaries infer latent attributes (e.g., identity or strategy

class) from trajectories (Hilprecht et al. (2019); Carlini et al. (2021); Pan et al. (2019)). Related work on adversarial robustness trains RL agents to resist perturbations (Pinto et al. (2017); Gleave et al. (2020)), but assumes active attackers degrading performance rather than passive observers inferring attributes. PALADIN's use of a co-trained adversary resembles adversarial training frameworks, including GAN-based privacy learning (Goodfellow et al. (2014a)) and DP self-play (Qiao & Wang (2024)), but differs by targeting single-agent trajectory privacy, embedding the adversary in the reward loop, and stabilising training via curriculum-guided penalties. Unlike adversarial debiasing, which suppresses correlations in supervised tasks, PALADIN operates in sequential settings and suppresses inference from full trajectories.

Another relevant line of work involves information-theoretic regularisation, such as the variational information bottleneck (Alemi et al. (2017); Achille & Soatto (2018)), and its extensions to privacy-preserving learning (Zhao et al. (2020); Noorbakhsh et al. (2024)). These methods provide representation-level guarantees but are less applicable to dynamic RL, where leakage arises from temporal dependencies in sequences. PALADIN complements these approaches by addressing the trajectory-level leakage that persists even after DP or robustness defences. Unlike static DP noise injection, PALADIN adapts its privacy shaping online via an adversary's inference success, using a curriculum to balance task mastery with progressive hardening against leakage.

To our knowledge, PALADIN is the first to integrate online adversarial inference, curriculum-guided shaping, and policy training in a unified RL framework. It reframes privacy as a proactive, adversary-aware signal within optimisation, rather than a rigid external constraint. While it does not provide worst-case $(\varepsilon, \delta)$-DP or information-theoretic bounds, PALADIN offers an empirical mechanism suited to settings where such guarantees are infeasible, for instance when sensitive variables are latent and unsupervised. Behavioural leakage thus remains a critical risk even when training-time DP or robustness defences are available, and it provides a practical, deployment-oriented mitigation.

# 3 METHODOLOGY

In this section, we introduce **PALADIN**, a proactive privacy–shaping framework that trains RL agents to suppress semantic leakage from their emitted behaviour while maintaining task performance. The key idea is to embed a co-trained adversary into the agent's reward loop, enabling privacy shaping as an explicit optimisation objective. We first formalise the privacy threat (Section 3.1), describe PALADIN's architecture and training loop (Section 3.2), and present a theoretical bound on adversarial inference (Section 3.3).

## 3.1 THREAT MODEL

We consider a *deployment-time* privacy threat where an external adversary passively observes an agent's trajectory and attempts to infer a sensitive attribute $y \in \mathcal{Y}$ (e.g., identity or strategy class). The adversary has no access to model internals, only to the transformed observation sequence $\tilde{X} = (\tilde{x}_1, \ldots, \tilde{x}_T)$ emitted by a policy $\pi_\theta$ with perturbation $f_\phi$. Behavioural leakage is the adversary's prediction success, measured by $L_{\text{leak}} = \ell(h(\tilde{X}), y)$. For example, in AV $y$ can indicate GPS spoofing status, while in FT, $y$ can denote a binary future price-movement label. In both cases, $y$ is inferred solely from trajectories.

This threat differs from *differential privacy*, which protects training data, and from *adversarial robustness*, which preserves task performance under perturbations. It also differs from adversarial debiasing and DP-RL, which perturb inputs, rewards, or gradients at training time. Prior work on GAN-based privacy or robustness focuses on representation obfuscation, but cannot prevent semantic inference from observable actions. PALADIN instead targets trajectory-level leakage, embedding a curriculum-guided privacy penalty directly into the reward loop.

**Problem Formulation:** We model the environment as an MDP $\mathcal{M} = (\mathcal{S}, \mathcal{A}, P, r, \gamma)$ with states $s_t$, actions $a_t$, and rewards $r(s_t, a_t)$. Observations $x_t \sim \mathcal{O}(s_t)$ are transformed as $\tilde{x}_t = f_\phi(x_t)$ before being consumed by the policy $\pi_\theta$. The objective is to maximise expected return while bounding leakage:

$$\max_{\theta,\phi} \mathbb{E}_\tau \left[ \sum_{t=1}^{T} \gamma^t r(s_t, a_t) \right] \quad \text{s.t.} \quad \mathbb{E}[L_{\text{leak}}(\tilde{X}, y)] \leq \varepsilon. \tag{1}$$

The term $\mathbb{E}_\tau \left[ \sum_{t=1}^{T} \gamma^t r(s_t, a_t) \right]$ in equation 1 is the standard RL utility (expected discounted return); the leakage constraint acts purely as a side condition.

We approximate this constrained problem with leakage-aware reward shaping:

$$r_t^{\text{total}} = r(s_t, a_t) - \lambda_t \, \widehat{L}(\tilde{X}_{1:t}), \tag{2}$$

where $\widehat{L}$ is a differentiable leakage proxy and $\lambda_t$ is a curriculum-driven penalty. This shaping term $-\lambda_t \, \widehat{L}(\tilde{X}_{1:t})$ is an explicit approximation. The surrogate leakage proxy does not perfectly represent a worst-case attacker, but provides a directional signal that penalises trajectories which are easy to classify. As with other learned reward-shaping components in RL, this biases the policy away from the unconstrained reward optimum, but here this bias is intentional since the goal is to trade some nominal return for reduced behavioural leakage. To avoid overfitting to a single surrogate, we (i) delay strong penalties via a curriculum until the base task is mastered, and (ii) evaluate all agents using separate held-out adversaries, often with mismatched architectures, that are never used in reward shaping. We do not claim formal $(\varepsilon, \delta)$-DP guarantees; instead, we validate the effectiveness of this shaping empirically using held-out adversaries with mismatched architectures. In practice, we instantiate $\widehat{L}$ using a parametrised adversary network $g_w(\tilde{X}_{1:t})$ trained to predict the sensitive label $y$ from transformed trajectories and set $\widehat{L}(\tilde{X}_{1:t})$ to its maximum posterior probability $\ell_t = \max_c \text{softmax}(o)_c$. The architecture and training procedure are detailed in Section 3.2. In all experiments, the terminal penalty level $\lambda_K$ is chosen via an initial $\lambda$-sweep (App F.1 and F), selecting the smallest value that yields a favourable privacy–utility trade-off without collapsing task return.

## 3.2 PALADIN FRAMEWORK

PALADIN is a closed-loop feedback mechanism in which a RL agent learns not only to maximise its task reward but also to minimise the amount of sensitive information that an external observer could infer from its behaviour. This is achieved through four interlocking modules discussed below:

**Transformation Network:** Each raw observation $x_t \in \mathbb{R}^d$ is processed by a neural network $f_\phi$, which learns to generate privacy-preserving perturbations. Concretely, we apply: $h_t = \text{ReLU}(W_1 x_t + b_1)$, $z_t = W_2 h_t + b_2$, $\tilde{x}_t = x_t + z_t$. where, $W_1$ and $W_2$ are learned weight matrices of dimensions $k \times d$ and $d \times k$ respectively. The residual connection $(x_t + z_t)$ preserves critical information for task performance while allowing the network to obfuscate sensitive features.

**Surrogate Leakage Predictor:** We then introduce a surrogate adversarial predictor $g_w$ to measure how much sensitive information remains in the transformed sequence $\tilde{X} = (\tilde{x}_1, \dots, \tilde{x}_t)$. This network attempts to infer the sensitive attribute $y$, providing a real-time proxy for adversarial inference capability. It processes recent observations via flattened sequences and layer-normalised projections, outputting class scores $o \in \mathbb{R}^C$. Thus, the leakage loss is computed as:

$$\mathcal{L}_{\text{leak}}(w) = -\sum_{c=1}^{C} \mathbf{1}\{y = c\} \log\big[\text{softmax}(o)\big]_c, \tag{3}$$

updating $w$ after each episode so that the leakage predictor $g_w$ remains a strong proxy for the true adversary.

**Curriculum-Guided Reward Shaping:** We wrap the environment's native reward $r_t$ in a custom Gym interface to incorporate privacy shaping. At each step, the wrapper performs the following tasks: i)Buffers the last $T$ transformed observations $\tilde{X}_{1:t}$. ii) Queries the leakage predictor $g_w$ to obtain its maximum posterior probability ($\ell_t = \max_c \text{softmax}(o)_c$). iii) Computes a shaped reward

$$r_t^{\text{shaped}} = r_t - \lambda_t \ell_t, \tag{4}$$

where $\lambda_t$ is a curriculum penalty weight that increases over training to gradually harden privacy constraints. Finally, tracks the cumulative difference $\sum_t \|\tilde{x}_t - x_t\|^2$ as a fidelity diagnostic to monitor perturbation magnitudes.

The curriculum design allows the agent to first focus on task mastery (by starting with $\lambda_t = 0$) before gradually adapting its policy to withstand adversarial inference.

**RL Agent:** We then employ standard off-the-shelf RL (e.g.Proximal Policy Optimisation (PPO) (PPO)), modified so that the agent receives the transformed observations $\tilde{x}_t$ directly as input to its

policy $pi_\theta(a \mid \tilde{x})$ and value functions. In both cases, we inject $f_\phi$ as a custom feature extractor so that the agent's policy $\pi_\theta(a \mid \tilde{x})$ and value or critic networks receive the transformed, privacy-shaped observations directly. Then, gradient updates flow simultaneously through the policy parameters $\theta$ and the transformation parameters $\phi$, thereby coupling task performance and privacy objectives. This closed-loop design embeds an adversary *in the training process* itself, offering an adaptive, data-driven defence: as the agent becomes more private, the adversary retrain to find new leakage, and the agent responds in turn. By integrating each component tightly and training them jointly, PALADIN transforms privacy from a post-hoc consideration into a core, first-class objective of the learning process. This design will let agents learn polices that perform the task while actively reducing the information they reveal in real time. We summarise PALADIN's joint training loop in Algorithm 1, and our architectural pipeline can be visualised in Figure 1 (–see App A).

---

**Algorithm 1** PALADIN: Joint Privacy-Shaping RL

---

**Require:** curriculum $(t_i, \lambda_i)_{i=0}^K$, predictor init $w$
 1: Initialize policy parameters $\theta$, transformer $\phi$, predictor $w$
 2: **for** episode = 1 **to** M **do**
 3:     Set $\lambda = \lambda_{\max\{i:t_i \leq \text{episode}\}}$
 4:     Collect rollout $(s_t, a_t, r_t)$ with observations $\tilde{x}_t = f_\phi(x_t)$
 5:     Compute $\widehat{L}(w; \tilde{X}, y)$
 6:     Form shaped rewards via equation 4
 7:     Update $(\theta, \phi)$ by RL algorithm on shaped rewards
 8:     Update predictor $w$ via $\nabla_w \widehat{L}$
 9: **end for**

---

### 3.3 THEORETICAL PRIVACY GUARANTEE

While PALADIN does not offer worst-case DP, we provide a simple bound that links task performance and leakage suppression. We emphasise that this bound is *not* a formal $(\varepsilon, \delta)$-DP guarantee. It relies on standard simplifying assumptions about the surrogate adversary (e.g., that a well-trained surrogate provides a faithful leakage proxy), which may not strictly hold in practice if the adversary underfits or if the data distribution shifts. The result should therefore be read as a **conceptual justification for our reward-shaping scheme**, rather than as a formal worst-case privacy guarantee.

**Theorem 1** (Privacy–Utility Bound). *Let $\widehat{L}(\tilde{X}, y) = -\log p_w(y \mid \tilde{X})$ denote the adversarial loss of a surrogate adversary with variance $\sigma^2$, and let the total reward $R = \sum_t r_t$ be bounded by $R_{\max}$. Then, under a final penalty coefficient $\lambda_K$, the expected adversarial loss is lower-bounded by $\mathbb{E}[\widehat{L}(\tilde{X}, y)] \geq \frac{\mathbb{E}[R] - \epsilon R_{\max}}{\lambda_K} - \sigma$, where $\epsilon$ captures residual convergence error.*

This bound is loose but interpretable as it expresses a trade-off between task reward $\mathbb{E}[R]$, the strength of the privacy penalty $\lambda_K$, and the expected error of an adversary evaluated on perturbed trajectories. Intuitively, policies that achieve very high reward under a strong leakage penalty cannot drive the adversarial loss arbitrarily low at the same time. The full proof is provided in App B. We therefore stress that Section 3.3 as a conceptual explanation of why a leakage-penalised reward can trace out a privacy–utility frontier, not as a formal $(\varepsilon, \delta)$-DP guarantee. All substantive privacy claims in this paper are empirical, based on evaluation with held-out adversaries and mismatched attack architectures.

## 4 EXPERIMENT

We empirically validate PALADIN's ability to balance privacy and utility across multiple domains. Here, we discuss our experiment based on the AV related dataset, and we detail the Finance sector in App F. We start by setting up benchmarks and then provide information on baselines.

## 4.1 BENCHMARK SETUP

We use the AV–GPS dataset Abrar et al. (2024), a collection of GPS traces from the ACL-Rover at the University of Arizona, with each point labelled as normal (**N**) or spoofed/attacked (**A**). The data comprise four subsets with varied spoofing conditions and 44 features per timestep; in total we obtain 115,855 samples. Full per-subset statistics and feature descriptions are given in App D. We impute missing values with the empirical mean, apply feature-wise min–max normalisation, and segment trajectories into fixed-length windows of $T = 100$ steps. Each window initialises an episode in which the recorded GPS trace is replayed, while the agent controls an additive perturbation to the *observable* GPS stream. Thus, the underlying physical trajectory is fixed, but the signal seen by an external observer can be reshaped by the agent.

Each 100-step GPS segment in our dataset is labelled **N** or **A** (spoofed/jammed). We treat this spoofing label as a *sensitive attribute from the perspective of an external observer*: a malicious eavesdropper or competitor could analyse the vehicle's publicly shared GPS stream to infer when spoofing attacks occur, which may reveal proprietary testing regimes or vulnerabilities. Our goal is not to hide attacks from trusted safety or regulatory monitors, which we assume continue to receive high-fidelity telemetry through authenticated channels, but to reduce what an unauthenticated third party can infer *purely* from the observable trajectory. PALADIN therefore applies its transformation network $f_\phi$ only to the externally visible GPS stream and shapes the agent's observable behaviour so that trajectories corresponding to normal and spoofed runs become harder to distinguish for a wide range of attack models, while the vehicle continues to navigate efficiently. In practice, this means driving down adversarial confidence and discrimination metrics (e.g. F1, AUROC) for held-out classifiers operating on on-policy trajectories, rather than guaranteeing perfectly random-guess performance. We model our environment as a finite Markov Decision Process (MDP) (– see Section 3) and restate here that our goal is to learn policy parameters $\theta$ and transformation parameters $\phi$ that jointly maximise cumulative reward while bounding information leakage (–see Equation (1)). The reward encourages "smooth movement" in the perturbed observation space, with a small quadratic control cost on the agent's perturbations.

**Adversarial Leakage Estimation:** To quantify behavioural privacy leakage, we train an auxiliary adversary $h_\phi$ to classify trajectories as normal or attacked. Our framework supports **multiple adversary architectures** to avoid overfitting conclusions to a single model class including i) **MLP Adversary.** A two-layer multilayer perceptron with hidden size 256, operating on flattened trajectory vectors. This baseline provides a strong but memoryless classifier. ii) **GRU Adversary.** A Gated Recurrent Unit (GRU) network that ingests trajectory sequences step by step, with hidden dimensions of $d_h = 128$. The GRU captures temporal dependencies and dynamics across the 100-step window. The final hidden state is mapped to class logits via a linear layer. iii) **Transformer Adversary.** A Transformer encoder with $L = 2$ layers and $H = 4$ attention heads, preceded by a linear embedding of the per-step features. Sequence representations are aggregated by mean pooling across time and projected to class logits. This model captures long-range dependencies and non-local correlations in the trajectory. Formally, given a trajectory $\tau_{1:T}$, the adversary outputs logits $h_\phi(\tau_{1:T})$ and is trained with cross-entropy loss: $\mathcal{L}_{\text{priv}} = -\frac{1}{M} \sum_{i=1}^{M} \log p_\phi\left(\widehat{y}^{(i)} = y_{\text{true}}^{(i)} \mid \tau_{1:T}^{(i)}\right)$ Parameters $\phi$ are optimised with ADAM ($10^{-3}$ learning rate) after each rollout. Crucially, the adversary used for training (the in-loop surrogate) is not the one used for reporting privacy metrics. After training each agent (baseline, static noise, DP-RL, DP-NASH, PALADIN), we generate on-policy trajectories and evaluate behavioural leakage using *separate, held-out* adversaries (MLP, GRU, Transformer) trained offline on disjoint data. For each configuration, we report Attack Accuracy, F1 score, and AUROC against these held-out attackers. This protocol directly addresses the risk of overfitting to a single surrogate adversary and provides a stricter test of PALADIN's privacy–utility trade-off. The environment is wrapped with a privacy-reward mechanism. At each step, the wrapper adjusts the task reward with a leakage penalty and a fidelity term: $r'_t = r_t - \lambda_t \ell_{\text{priv}} - \beta \operatorname{MSE}(\widetilde{\mathbf{s}}_t, \mathbf{s}_t^{\text{raw}})$, where $\ell_{\text{priv}}$ is either adversary confidence ($\max_k \operatorname{softmax}$) or negative log-likelihood of the true class, and $\beta$ controls how strongly we penalise large perturbations of the observable signal.

The curriculum for $\lambda_t$ follows staged increases, e.g. $(0, 0.0)$, $(200, 0.5)$, $(400, 1.0)$, $(600, 2.0)$, $(800, 4.0)$, allowing the agent to first learn the navigation task before progressively enforcing privacy. For all AV–GPS results, privacy metrics are computed *post hoc* from on-policy trajectories rolled out by the trained agents and evaluated with held-out adversaries as described above. This separates the

shaping signal used during training from the attack models used for evaluation. To evaluate PALADIN beyond autonomous navigation, we construct analogous environments in the financial domain; details are provided in App D.2 and E. In this case, we use a deterministic replay environment over historical OHLCV windows, treating the task as a sequential decision problem on fixed trajectories rather than a full market-impact simulator. This isolates deployment-time behavioural leakage from the orthogonal challenge of modelling interactive price dynamics.

In terms of evaluation metric, we used several off-the-shelf metrics such as F1, AUROC, but our main evaluation metrics are leak_conf and leak_nll which we will briefly explain below: Formally, let $p_\psi(y \mid \tau)$ denote the prediction of a held-out adversary on a trajectory $\tau$, and $y^\star$ its true sensitive label. We define *leak_conf* as the adversary's confidence in the true class, **leak_conf** $= \mathbb{E}_\tau\big[p_\psi(y^\star \mid \tau)\big]$, and *leak_nll* as the corresponding negative log-likelihood, leak_nll $= \mathbb{E}_\tau\big[-\log p_\psi(y^\star \mid \tau)\big]$. Lower leak_conf and higher leak_nll both indicate that the adversary assigns less probability to the true label, but they are not identical: leak_conf is sensitive to small shifts in the top-class probability, whereas leak_nll reflects changes in the full predictive distribution and penalises over-confident mistakes more strongly. We therefore report both metrics, and interpret them jointly with attack F1 and AUROC when assessing behavioural leakage.

## 4.2 BASELINES

We compare PALADIN against several baselines described below. Each baseline represents a distinct privacy–utility paradigm, allowing us to isolate the contribution of PALADIN. **Standard RL** employs a vanilla RL agent that optimises only for task performance, with no mechanism to mitigate privacy leakage. Formally, the policy parameters $\theta$ are chosen to maximise the expected discounted return under the usual Markovian dynamics $P(s_{t+1} \mid s_t, a_t)$. By design, this agent attains an upper bound on utility but offers no protection against adversarial inference and serves as our "no privacy" reference.

$$J(\theta) = \mathbb{E}_{\tau \sim \pi_\theta}\Big[\sum_{t=1}^{T} \gamma^{t-1} r(s_t, a_t)\Big], \tag{5}$$

**Static Noise Injection** injects Gaussian noise with fixed variance $\sigma^2$ into the observation and action channels. In **Static_Obs**, each raw observation is perturbed as $\tilde{x}_t = x_t + \epsilon_t$, $\epsilon_t \sim \mathcal{N}(0, \sigma^2 I)$, whereas in **Static_Act**, the executed action is noisy: $\tilde{a}_t = a_t + \epsilon_t$, $\epsilon_t \sim \mathcal{N}(0, \sigma^2 I)$. These methods are easy to implement and incur minimal overhead, but their non-adaptive nature often forces a trade-off: too much noise degrades utility severely, while too little fails to impede adversarial inference.

**Adversarial Shaping without Curriculum (Adv_No_Cur)** Here we integrate a surrogate adversary $h$ into the reward but hold the privacy penalty $\lambda$ fixed throughout training. At each time step, the shaped reward becomes $r_t^{\text{shaped}} = r(s_t, a_t) - \lambda\, \ell\big(h(\tilde{X}_{1:t}), y\big)$, where $\ell$ is the cross-entropy loss of the adversary's prediction of the sensitive label $y$. By co-training policy and adversary, this baseline encourages privacy-aware behaviour, but the constant penalty can either hamper early task learning (if $\lambda$ is large) or fail to enforce privacy (if $\lambda$ is small). In our experiments, Adv_No_Cur uses a fixed, relatively small privacy weight $\lambda$ chosen from the $\lambda$-sweep as the largest value that keeps training stable for a given train–test pairing. This configuration is therefore a tuned, fixed-penalty baseline and should not be confused with the No Curriculum ablation in Section 5, which applies the much stronger PALADIN terminal weight $\lambda_K$ from the beginning of training.

**DP-RL** This baseline enforces formal $(\varepsilon, \delta)$-differential privacy on the policy parameters using the DP-SGD mechanism of (Abadi et al. (2016)). For each minibatch of size $L$, we compute per-sample gradients $g_i = \nabla_\theta \mathcal{L}_i$, clip each to norm $C$, and aggregate with Gaussian noise:

$$\bar{g} = \frac{1}{L} \sum_{i=1}^{L} \text{clip}(g_i, C) + \mathcal{N}\big(0, \sigma^2 C^2 I\big). \tag{6}$$

This update guarantees $(\varepsilon, \delta)$-DP, where $\varepsilon = f(\sigma, C, L, T)$ for $T$ total steps. While DP-RL prevents model-inversion attacks on the trained parameters, it does not constrain the agent's observable behaviours: the actions and trajectories executed at test time may still reveal sensitive information. This highlights the need for methods such as PALADIN that shape privacy in the observation space and the agent's behaviour, rather than only the parameters.

**DP-NASH** extends the DP-RL baseline, influenced by (Qiao & Wang (2024)), where we incorporate adversarially guided optimisation, forming a min-max training loop between the RL agent and a leakage-predicting adversary. The RL agent aims to maximise task performance, while the adversary is trained to infer sensitive attributes from the agent's behavioural traces. The combined optimisation objective is formalised as:

$$\min_{\theta} \max_{\phi} \; \mathbb{E}_{\tau \sim \pi_{\theta}} \left[ \sum_{t=1}^{T} \gamma^{t-1} \left( r(s_t, a_t) - \lambda \, \ell\big(h_{\phi}(\tilde{X}_{1:t}), y\big) \right) \right], \tag{7}$$

where $\theta$ denotes the policy parameters, $\phi$ denotes the adversary parameters, $h_{\phi}$ is the adversary network predicting sensitive label $y$ based on trajectory prefix $\tilde{X}_{1:t}$, and $\ell$ is the leakage loss, typically cross-entropy. While this formulation suggests a joint min-max optimisation, in practice, we alternate training steps: the policy is optimised solely for task performance (standard PPO objective), and the adversary is trained separately to maximise leakage prediction accuracy using the latest trajectories. This setup enables indirect adversarial shaping: although the policy does not explicitly minimise leakage during updates, its behaviour is influenced over time by the adversary's evolving strength. To ensure formal privacy protection, we apply DP-SGD to both the policy and adversary models. Specifically, per-sample gradients are clipped and perturbed with Gaussian noise:

$$g_i^{\text{policy}} = \nabla_{\theta} \mathcal{L}_i^{\text{RL}}, \quad \bar{g}^{\text{policy}} = \frac{1}{L} \sum_{i=1}^{L} \text{clip}(g_i^{\text{policy}}, C) + \mathcal{N}(0, \sigma^2 C^2 I), \tag{8}$$

$$g_i^{\text{adv}} = \nabla_{\phi} \, \ell\big(h_{\phi}(\tilde{X}_{1:T}), y\big), \quad \bar{g}^{\text{adv}} = \frac{1}{L} \sum_{i=1}^{L} \text{clip}(g_i^{\text{adv}}, C) + \mathcal{N}(0, \sigma^2 C^2 I), \tag{9}$$

where $L$ is the minibatch size, $C$ is the gradient clipping threshold, and $\sigma$ controls the noise magnitude. This dual-DP mechanism provides that both the agent's policy and the adversary's parameters satisfy $(\varepsilon, \delta)$, DP guarantees. Further discussion is detailed in App C.

## 5 EXPERIMENTAL RESULT AND DISCUSSION

We evaluate PALADIN against baseline and privacy-aware RL approaches on the AV-GPS dataset. Our analysis focuses on the trade-off between utility (measured by return) and privacy leakage, where leakage is quantified using adversary confidence (`leak_conf`), negative log-likelihood (`leak_nll`), and attack metrics (accuracy, F1, and AUROC; Att Acc, Att F1, Att A in Table 1).

Table 1 reports results for five representative train–test adversary pairings. For **MLP–GRU**, the baseline agent achieves moderate utility (return 16.14) with non-trivial leakage (`leak_conf`= 0.80, Attack F1 = 0.46, Att Acc = 0.86, AUROC ≈ 0.68). Static noise is surprisingly competitive here: `static_obs` and `static_act` slightly improve return (21.30 and 24.31) and keep the attack metrics in a similar range (Acc ≈ 0.86, F1 ≈ 0.47, AUROC ≈ 0.67–0.68). The adversarial variant without curriculum (`adv_no_cur`) yields a small privacy improvement (F1 = 0.44) and modest return (19.77). DP baselines (DP-RL, DP-Nash) reduce return to single digits while only marginally altering Attack F1. PALADIN achieves the strongest overall trade-off: it substantially increases return to 30.56, almost double the baseline, while maintaining leakage at a comparable level (Att Acc = 0.86, Att F1 = 0.45, AUROC ≈ 0.67, `leak_conf`= 0.77). In this pairing, PALADIN does not minimise leakage absolutely, but it moves the privacy–utility frontier outward. In terms of the **MLP–Transformer** setting, PALADIN provides the clearest benefit, where baseline and most baselines exhibit strong leakage (Att Acc ≥ 0.89, Att F1 ≥ 0.86, AUROC ≈ 0.95–0.99, `leak_conf`≥ 0.97). `static_obs` marginally improves privacy (F1 = 0.86) and utility (return 34.23), but remains highly leaky. PALADIN, in contrast, simultaneously maximises utility and sharply reduces leakage: return increases to 40.90, while Att Acc and Att F1 drop to 0.16 and 0.14 respectively and `leak_conf` falls to 0.14, with a corresponding increase in `leak_nll`; AUROC remains high (Att A = 1.00), indicating that the adversary's ranking ability is preserved even though its default decision rule becomes almost useless. No static or DP baseline approaches this pairing, confirming that proactive shaping can meaningfully distort behavioural signatures even against a powerful Transformer adversary. Likewise, for the **GRU–Transformer**, the baseline is highly exposed (Att Acc = 1.00, Att F1 = 1.00, Att A = 1.00, `leak_conf`= 0.997) and has

Table 1: Experimental results on AV-GPS. where Att = Attack and Att A=Attack AUROC

| Train/Test pair | Method | Return | $\sigma_{\text{return}}$ | leak_conf | $\sigma_{\text{conf}}$ | leak_nll | $\sigma_{\text{nll}}$ | Att Acc | Att F1 | Att A |
|---|---|---|---|---|---|---|---|---|---|---|
| MLP–MLP | baseline | 19.85 | 3.72 | 0.998 | 0.001 | 0.002 | 0.001 | 1.000 | 1.000 | 1.000 |
| | static_obs | 31.55 | 3.11 | 0.595 | 0.278 | 0.688 | 0.719 | 0.580 | 0.525 | 0.884 |
| | static_act | 27.43 | 5.23 | 0.933 | 0.139 | 0.162 | 0.783 | 0.990 | 1.000 | 1.000 |
| | adv_no_cur | 40.28 | 4.85 | 0.996 | 0.012 | 0.004 | 0.013 | 0.980 | 0.932 | 1.000 |
| | dp_rl | 9.13 | 0.75 | 0.994 | 0.015 | 0.006 | 0.016 | 0.950 | 0.942 | 0.967 |
| | dp_nash | 12.73 | 0.60 | 0.998 | 0.001 | 0.002 | 0.001 | 0.990 | 1.000 | 1.000 |
| | PALADIN | 38.30 | 4.00 | 0.994 | 0.002 | 0.006 | 0.002 | 1.000 | 1.000 | 1.000 |
| MLP–GRU | baseline | 16.14 | 5.43 | 0.795 | 0.238 | 0.337 | 0.575 | 0.860 | 0.462 | 0.680 |
| | static_obs | 21.30 | 17.10 | 0.739 | 0.296 | 0.479 | 0.719 | 0.860 | 0.468 | 0.679 |
| | static_act | 24.31 | 9.39 | 0.808 | 0.211 | 0.290 | 0.483 | 0.860 | 0.468 | 0.670 |
| | adv_no_cur | 19.77 | 6.19 | 0.733 | 0.286 | 0.472 | 0.682 | 0.850 | 0.444 | 0.682 |
| | dp_rl | 9.68 | 1.15 | 0.823 | 0.196 | 0.264 | 0.464 | 0.850 | 0.432 | 0.677 |
| | dp_nash | 8.71 | 1.08 | 0.722 | 0.303 | 0.512 | 0.729 | 0.860 | 0.468 | 0.661 |
| | PALADIN | 30.56 | 2.07 | 0.765 | 0.240 | 0.377 | 0.566 | 0.860 | 0.451 | 0.669 |
| MLP–Transformer | baseline | 25.26 | 17.70 | 0.996 | 0.002 | 0.005 | 0.002 | 0.990 | 0.956 | 0.987 |
| | static_obs | 34.23 | 1.89 | 0.980 | 0.074 | 0.025 | 0.105 | 0.890 | 0.864 | 0.947 |
| | static_act | 26.58 | 10.89 | 0.973 | 0.140 | 0.159 | 1.059 | 0.960 | 0.968 | 0.984 |
| | adv_no_cur | 27.72 | 1.40 | 0.992 | 0.013 | 0.008 | 0.013 | 0.990 | 1.000 | 0.988 |
| | dp_rl | 13.43 | 2.19 | 0.996 | 0.002 | 0.004 | 0.002 | 1.000 | 1.000 | 0.989 |
| | dp_nash | 8.95 | 1.22 | 0.987 | 0.048 | 0.014 | 0.059 | 0.980 | 1.000 | 0.980 |
| | PALADIN | 40.90 | 2.45 | 0.140 | 0.347 | 8.350 | 3.370 | 0.160 | 0.138 | 1.000 |
| Transformer–GRU | baseline | 24.13 | 6.32 | 0.989 | 0.008 | 0.011 | 0.008 | 1.000 | 1.000 | 0.997 |
| | static_obs | 29.67 | 4.32 | 0.987 | 0.003 | 0.013 | 0.003 | 1.000 | 0.864 | 0.984 |
| | static_act | 31.50 | 4.64 | 0.972 | 0.126 | 0.057 | 0.332 | 0.980 | 0.956 | 0.974 |
| | adv_no_cur | 21.40 | 2.35 | 0.991 | 0.004 | 0.009 | 0.004 | 0.990 | 1.000 | 0.993 |
| | dp_rl | 8.83 | 0.74 | 0.968 | 0.137 | 0.100 | 0.608 | 0.950 | 1.000 | 0.968 |
| | dp_nash | 7.74 | 0.26 | 0.987 | 0.005 | 0.013 | 0.005 | 0.990 | 0.965 | 0.988 |
| | PALADIN | 25.02 | 1.51 | 0.991 | 0.008 | 0.009 | 0.008 | 0.990 | 0.961 | 0.997 |
| GRU–Transformer | baseline | 16.33 | 8.04 | 0.997 | 0.001 | 0.003 | 0.001 | 1.000 | 1.000 | 1.000 |
| | static_obs | 23.99 | 4.47 | 0.985 | 0.059 | 0.017 | 0.075 | 0.980 | 0.956 | 1.000 |
| | static_act | 16.85 | 8.23 | 0.983 | 0.090 | 0.025 | 0.145 | 1.000 | 1.000 | 1.000 |
| | adv_no_cur | 20.52 | 6.64 | 0.995 | 0.007 | 0.005 | 0.008 | 0.940 | 0.903 | 0.978 |
| | dp_rl | 6.76 | 0.97 | 0.983 | 0.090 | 0.025 | 0.146 | 0.940 | 0.910 | 0.956 |
| | dp_nash | 9.05 | 1.26 | 0.989 | 0.034 | 0.012 | 0.038 | 1.000 | 1.000 | 1.000 |
| | PALADIN | 18.01 | 1.95 | 0.582 | 0.303 | 0.751 | 0.739 | 0.580 | 0.559 | 1.000 |

relatively low return (16.33). Static observation noise improves both return and privacy (return 23.99, Attack F1 = 0.96), and adv_no_cur offers a small privacy gain (F1 = 0.90) with return 20.52. PALADIN occupies a distinct region of the trade-off: it improves return over baseline to 18.01 and, crucially, yields the lowest Att Acc and Att F1 (0.58 and ≈ 0.56 respectively) and markedly lower leak_conf (0.58), while AUROC remains high (Att A = 1.00). Although static baselines can achieve higher returns, PALADIN is the only method that achieves a strong reduction in both adversary confidence and Attack F1, making it our primary "high-privacy" configuration for this pairing. In terms of the **Transformer–GRU** pairing, which illustrates a complementary pattern. Baseline RL already yields good utility (return 24.13) but is fully exposed (Att Acc = 1.00, Att F1 = 1.00, Att A ≈ 0.997, leak_conf ≈ 0.99). Here, static_obs is a strong baseline, improving return to 29.67 and reducing Att F1 to 0.86 (with Att Acc still = 1.00 and AUROC ≈ 0.98) alongside a slight reduction in leak_conf. PALADIN gives a modest utility gain over baseline (return 25.02) and a small privacy improvement (Att F1 = 0.96, AUROC unchanged at ≈ 0.997), but does not outperform static_obs on either axis. This case highlights that simple obfuscation strategies can be competitive when the adversary architecture is comparatively weak. Finally, the **MLP–MLP** pairing exhibits that PALADIN primarily acts as a utility booster. The baseline has return 19.85 with near-perfect leakage (Att Acc = 1.00, Att F1 = 1.00, Att A = 1.00, leak_conf = 0.998). static_obs is again a strong privacy baseline, achieving the lowest Att F1 (0.53), lower Att Acc (0.58), reduced AUROC (≈ 0.88), and substantially lower leak_conf (0.60) while improving return to 31.55. Whereas PALADIN increases return further to 38.30 but keeps Att Acc and Att F1 at 1.00 and only slightly lowers leak_conf. Thus, for MLP–MLP, the main benefit is utility rather than privacy, and we treat it as a sanity check for training stability rather than a flagship privacy result. Across all pairings, DP-RL and DP-Nash consistently yield low returns (typically < 10) with only modest changes in leakage, highlighting the limitations of parameter-level DP for controlling behavioural signals.

**Effect of the privacy weight $\lambda$:** We study the impact of the privacy penalty by sweeping $\lambda \in \{0, 0.1, 0.2, 0.5, 1, 2\}$ for the selected seed in each pairing (Table 3). Across all settings, the resulting curves are non-monotonic in both utility and privacy, where very small $\lambda$ under-enforce privacy in the high-leak pairings, while very large $\lambda$ often harm return without providing additional leakage reduction. For MLP–GRU, moderate penalties ($\lambda \in \{0.1, 0.5\}$) provide the best trade-offs, yielding returns substantially above baseline while keeping the attack metrics (Acc, F1, AUROC) in a similar range to $\lambda = 0$. For GRU–Transformer, $\lambda = 0.5$ achieves a clear privacy–utility improvement, simultaneously increasing return relative to baseline and reducing both adversary confidence and Attack F1, with AUROC dropping from 1.0 to $\approx 0.86$. Overall, these sweeps indicate that PALADIN is robust across a range of $\lambda$ values but benefits most from moderate, curriculum-guided privacy weights; full per-pair results are deferred to the supplementary material (App.Table 3).

**Ablation study:** We next ablate PALADIN's main components on the same pairings (Table 4). We focus on three design elements: the learned transformation network (`no_transform`, `shallow_phi`), curriculum scheduling (`no_curriculum`), and adversary capacity (`small_adv`, `weak_adv`), and compare them with DP baselines (DP-RL and DP-NASH). Across all pairings, *removing the curriculum* leads to severe instability where `no_curriculum` variants exhibit highly negative or extremely high-variance returns while leaving the attack metrics essentially in their high-leak regime (Acc and F1 near 1.0, AUROC close to 1.0), confirming that strong privacy penalties applied from the outset prevent the agent from learning useful behaviour. While variants without an expressive transformation (`no_transform`, `shallow_phi`) usually retain moderate returns, in the challenging settings (MLP–GRU, MLP–Transformer, Transformer–GRU and GRU–Transformer) they fail to meaningfully reduce adversary confidence or attack performance (Acc, F1, AUROC), indicating that representation learning is necessary to decouple task-relevant and sensitive features. The MLP–MLP `shallow_phi` outlier achieves strong privacy (low `leak_conf`, reduced Acc/F1, AUROC $\approx 0.60$) but in a relatively simple backbone pairing that we treat primarily as a training sanity check. On the other hand, weakening the adversary (`small_adv`, `weak_adv`) generally preserves or improves return but leaves the agent highly leaky, with Acc, F1, and AUROC remaining near 1.0, as the policy no longer receives informative privacy gradients. DP-RL and DP-NASH again yield low utility with only modest leakage reductions, echoing our core results. Overall, these ablations support our claim that PALADIN's effectiveness arises from the *combination* of an expressive transformation network, curriculum-scheduled privacy penalties, and a sufficiently strong adversary; detailed per-pair ablation statistics are reported in the supplementary material (App F.2).

# 6 Conclusion and Future Work

We introduced PALADIN, a proactive privacy–shaping framework that embeds an adversarial leakage estimator directly into the RL reward loop. By co-training a surrogate adversary with curriculum-scheduled penalties, PALADIN achieves strong task returns while reducing deployment-time behavioural leakage on AV–GPS and financial-trading benchmarks, outperforming static noise injection and DP-based baselines. Ablation studies confirm that both an expressive transformation network and curriculum scheduling are indispensable: removing either destabilises training or leaves trajectories highly identifiable. Although PALADIN does not provide worst-case $(\varepsilon, \delta)$–DP guarantees, it offers a practical recipe for empirically controlling behavioural leakage, and could be combined with DP mechanisms or information-theoretic bounds to protect both parameters and trajectories. As immediate future work, we aim to integrate PALADIN with DP mechanisms to jointly protect training data and behavioural emissions, and to pursue formal guarantees via mutual-information-based analysis. Broader extensions include *multi-agent* settings where inter-agent communication introduces additional leakage channels, *adaptive adversaries* (e.g., recurrent or attention-based) that dynamically co-evolve with the policy, and scaling to *high-dimensional sensory inputs* such as vision or LiDAR. Our current evaluation is limited to an interactive navigation task (AV–GPS) and a deterministic-replay financial benchmark, where agent actions do not affect underlying market dynamics. Extending PALADIN to fully interactive, high-dimensional control domains, including real-time games, CARLA-style autonomous-driving simulators, and robot manipulation, and to realistic market simulators with endogenous price impact, alongside real-world deployment on physical platforms and edge devices to assess latency and energy constraints, are important directions for future work.PALADIN provides a principled mechanism to negotiate the privacy–utility trade-off by elevating privacy to a first-class training signal rather than a post-hoc add-on. This moves us closer to deploying RL agents safely in adversarial real-world environments.

ETHICS STATEMENT

This work does not involve human or animal subjects, nor does it use sensitive personal data. All datasets are publicly available and anonymised (AV-GPS and Yahoo Finance). The proposed framework is designed to *mitigate* privacy leakage rather than introduce new risks. We explicitly acknowledge that PALADIN does not provide formal $(\varepsilon, \delta)$–DP guarantees; instead, we report empirical privacy–utility trade-offs transparently and discuss the limitations of our approach. We identify no conflicts of interest or ethical concerns beyond those addressed in the paper. We also make use of a Large Language Model-based tool to assist with identifying related literature.

In the AV-GPS dataset, the sensitive attribute corresponds to whether a trajectory originates from a GPS-spoofed or normal driving segment. Our intention is not to conceal spoofing from internal safety monitors or regulatory audit systems. PALADIN acts only on the externally observable trajectory, modelling what an unauthorised third-party observer could infer from passively observing vehicle motion. Internal diagnostics, authenticated telemetry, and safety-critical subsystems operate on untransformed sensor streams and therefore remain unaffected by PALADIN. We emphasise this distinction to avoid conflating behavioural privacy with safety-critical anomaly masking.

REPRODUCIBILITY STATEMENT

We provide detailed descriptions of algorithms, architectures (MLP, GRU, Transformer), hyperparameters, and evaluation metrics in Section 4 and the Appendices D and F. Dataset preprocessing steps are documented ( Section 4 and App D), and pseudocode is included (Algorithm 1). To support replication, we will release source code and configuration files upon acceptance.

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

We present all the supplementary materials in the appendix.

# A   PALADIN PIPELINE

We present PALADIN Architecture in this section.

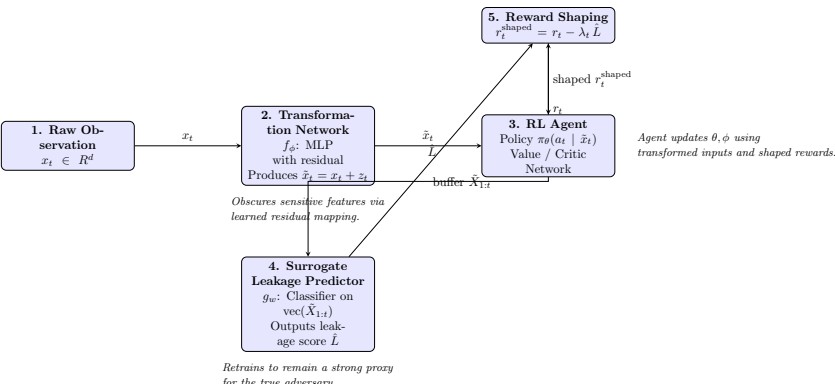

Figure 1: PALADIN Pipeline: observations pass through the transformation network $f_\phi$, the surrogate predictor $g_w$ estimates leakage, and the RL agent receives a penalised reward.

# B   DETAILED PROOF OF THEOREM

In this appendix, we give a more expansive, yet intuitive, derivation of the Privacy–Shield Bound from Theorem 1. Recall that at convergence, PALADIN optimises the shaped objective

$$J(\theta,\phi;w) \ = \ \mathbb{E}_{\tau\sim\pi_\theta}\Big[\sum_{t=1}^{T} r(s_t,a_t)\Big] \ - \ \lambda_K \, \mathbb{E}_{\tau\sim\pi_\theta}\big[\widehat{L}(w;\tilde{X},y)\big], \tag{10}$$

where $\widehat{L}(w;\tilde{X},y) = -\log p_w(y \mid \tilde{X})$ is our surrogate leakage loss ((cf. adversarial training in (Goodfellow et al. (2014b))). We discuss this in two steps below:

STEP 1: STATIONARITY OF THE JOINT OBJECTIVE

At convergence, PALADIN has jointly optimised the policy and transformation parameters $(\theta,\phi)$ under a fixed adversary $w$ (Sutton & Barto (2018)). Concretely, it maximises the shaped objective

$$J(\theta,\phi;w) = \underbrace{\mathbb{E}_{\tau\sim\pi_\theta}\Big[\sum_{t=1}^{T} r(s_t,a_t)\Big]}_{\mathbb{E}[R]} - \lambda_K \underbrace{\mathbb{E}_{\tau\sim\pi_\theta}\big[\widehat{L}(w;\tilde{X},y)\big]}_{\mathbb{E}[\widehat{L}]},$$

where $\widehat{L}(w;\tilde{X},y) = -\log p_w(y \mid \tilde{X})$ is our surrogate leakage loss and $\lambda_K$ is the final privacy penalty.

Since $(\theta,\phi)$ are (approximately) optimal, the gradient of $J$ with respect to these parameters vanishes:

$$\nabla_{\theta,\phi} J(\theta,\phi;w) \ \approx \ 0.$$

Intuitively, this means that any infinitesimal change in the policy or transformation that would increase the raw reward $\mathbb{E}[R]$ must be offset by an equal or greater increase in the expected leakage term $\lambda_K \mathbb{E}[\widehat{L}]$, and vice versa.

Rearranging the two competing terms yields

$$\lambda_K \mathbb{E}[\widehat{L}] \ \approx \ \mathbb{E}[R] - C,$$

where $C$ is a nonnegative slack term capturing residual suboptimality, arising from finite-step optimisation, approximation error (Kakade (2003)), and nonzero gradients in high-dimensional parameter spaces. Since each per-step reward satisfies $|r(s_t, a_t)| \leq r_{\max}$, the total raw return $R = \sum_{t=1}^{T} r(s_t, a_t)$ is bounded by $R_{\max} = T \, r_{\max}$. Thus, one may bound

$$0 \leq C \leq \epsilon \, R_{\max}, \quad \epsilon \geq 0,$$

where $\epsilon$ captures the fraction of the maximum return lost to optimisation slack.

Substituting this bound into the rearranged stationarity condition gives a lower bound on the expected surrogate leakage:

$$\mathbb{E}[\widehat{L}] \geq \frac{\mathbb{E}[R] - \epsilon \, R_{\max}}{\lambda_K}. \tag{11}$$

In words, at equilibrium, the privacy penalty $\lambda_K$ must be large enough that the expected leakage cannot fall below the gap between the achievable task return and any optimisation slack, scaled by $\lambda_K^{-1}$. This relation formalises the trade-off enforced by PALADIN's shaped objective: improving privacy (reducing $\mathbb{E}[\widehat{L}]$) necessarily incurs a cost in task return $\mathbb{E}[R]$, and vice versa.

### STEP 2: FROM SURROGATE TO TRUE LEAKAGE

Our analysis thus far bounds the *expected surrogate loss* $\mathbb{E}[\widehat{L}]$. To translate this into a guarantee on the *true adversary loss* $\ell(h(\tilde{X}), y)$, we invoke the unbiasedness and concentration properties of our surrogate estimator. By construction,

$$\mathbb{E}[\widehat{L}] = \mathbb{E}\big[\ell(h(\tilde{X}), y)\big],$$

and we assume a finite estimator variance $\mathrm{Var}[\widehat{L}] = \sigma^2$.

Applying Chebyshev's inequality, for any $\delta \in (0, 1)$,

$$\Pr\big[\, |\widehat{L} - \mathbb{E}[\widehat{L}]| \geq \sigma/\sqrt{\delta} \,\big] \leq \delta.$$

In other words, with probability at least $1 - \delta$, the realised surrogate loss lies within $\sigma/\sqrt{\delta}$ of its expectation. Equivalently, in expectation, one obtains

$$\mathbb{E}\big[\ell(h(\tilde{X}), y)\big] \geq \mathbb{E}[\widehat{L}] - \sigma.$$

Substituting the lower bound from equation 11 immediately yields

$$\mathbb{E}\big[\ell(h(\tilde{X}), y)\big] \geq \frac{\mathbb{E}[R] - \epsilon \, R_{\max}}{\lambda_K} - \sigma,$$

completing the derivation of the Privacy–Shield Bound in Theorem 1. This final inequality makes explicit the trade-off: achieving a high expected return $\mathbb{E}[R]$ under a given penalty $\lambda_K$ necessarily enforces a lower bound on the adversary's expected inference loss, up to the surrogate variance $\sigma$.

### B.1 DISCUSSION OF KEY ASSUMPTIONS

Before applying the Privacy–Shield Bound, we define several standard but crucial assumptions:

- **Surrogate Leakage Proxy:** We assume that the learned adversary $g_w$ provides a well-defined, differentiable proxy for the true leakage $\ell(h(\tilde{X}), y)$. In practice, we keep $g_w$ competitive by retraining it on fresh trajectories throughout training, so that its loss tracks changes in the policy's behavioural leakage.

- **Bounded Rewards:** Each per-step reward satisfies $|r(s_t, a_t)| \leq r_{\max}$, so the total return $R = \sum_{t=1}^{T} r(s_t, a_t)$ is finite. This is a standard requirement in finite-horizon RL analyses.

- **Convergence Slack ($\epsilon$):** Real-world optimisers only approximate stationarity. We introduce $\epsilon \geq 0$ to capture residual gradient magnitude or suboptimality at convergence. Empirically, $\epsilon$ is small once learning stabilises.

- **Concentration Inequality.** We apply Chebyshev's inequality to bound the deviation of $\widehat{L}$ from its mean. If one can assume sub-Gaussian tails for the surrogate loss, tighter high-probability guarantees follow from Bernstein bounds at the expense of stronger moment conditions.

With these assumptions, the bound is prepared

$$\mathbb{E}[\ell(h(\tilde{X}), y)] \geq \frac{\mathbb{E}[R] - \epsilon\, R_{\max}}{\lambda_K} - \sigma.$$

In particular, to enforce a minimum adversarial loss $\delta$, one selects

$$\lambda_K \geq \frac{\mathbb{E}[R] - \epsilon\, R_{\max}}{\delta + \sigma},$$

which clearly trades off utility ($\mathbb{E}[R]$) against privacy (controlled by $\delta$ and $\sigma$).

To reiterate, this bound is not a formal $(\varepsilon, \delta)$-DP guarantee. It relies on standard simplifying assumptions about the surrogate adversary (e.g., that a sufficiently well-trained $g_w$ provides a faithful leakage proxy), which may not strictly hold in practice if the adversary underfits or if the data distribution shifts. Hence, the bound should be interpreted as a conceptual explanation of how the shaped objective induces a privacy–utility trade-off, rather than as a worst-case privacy guarantee.

## C  EXTENDED DISCUSSION ON DP-NASH

DP-NASH operationalises privacy defence via two intertwined mechanisms: (i) formal DP guarantees at the parameter level, protecting against extraction or inversion attacks on model weights, and (ii) adversarial shaping, which pressures the policy to evolve privacy-preserving behaviours indirectly through buffer dynamics and adversarial retraining. This dynamic creates an arms-race-like environment, where the adversary continuously adapts to exploit emerging leakage patterns, and the policy gradually internalises strategies that reduce its observability footprint.

While DP-NASH is a powerful baseline, merging DP with adversarial privacy dynamics, it has practical trade-offs. The privacy penalty $\lambda$ is fixed throughout training, which may under- or over-penalise depending on the agent's learning stage. Additionally, as the adversary strengthens, training can become unstable, particularly when privacy noise is high, requiring careful tuning of hyperparameters to balance privacy, utility, and learning stability.

As mentioned in Section 4.2, DP-NASH is influenced by the (Qiao & Wang (2024)) framework, which implements Nash value iteration and equilibrium computation in a multi-agent Markov game setting. However, DP-NASH is designed specifically for single-agent RL with behavioural privacy concerns. Therefore, rather than reformulating the problem as a game between environment agents, it employs a dedicated leakage-predicting adversary within a standard RL pipeline, allowing us to emulate adversarial privacy dynamics in a single-agent context. This makes DP-NASH a rigorous and technically meaningful baseline to benchmark PALADIN's performance, while remaining fully aligned with the scope of behavioural privacy in standard RL tasks.

**Privacy Guarantee Sketch for DP-NASH**  DP-NASH applies DP-SGD independently to both the policy network and the adversarial leakage predictor. Each model satisfies $(\varepsilon, \delta)$-DP as guaranteed by the moments accountant (Abadi et al. (2016)). Specifically, for each model, the privacy guarantee is:

$$\varepsilon = f(\sigma, C, L, T),$$

where $\sigma$ is the noise multiplier, $C$ is the clipping norm, $L$ is the minibatch size, and $T$ is the total number of steps. Because both the policy and adversary access the same sensitive data (trajectories), standard composition of DP applies. By the basic composition theorem, the overall privacy budget satisfies

$$\varepsilon_{\text{total}} \leq \varepsilon_{\text{policy}} + \varepsilon_{\text{adv}}$$

for a common $\delta$. This guarantees that any observer of both the policy and adversary models learns at most $(\varepsilon_{\text{total}}, \delta)$-differentially private information about any individual trajectory.

Importantly, the alternating training schedule, where the policy and adversary are updated in different phases, does not break DP because the noise injection and privacy accounting are applied at each step independently, and DP is preserved under post-processing. Thus, DP-NASH's privacy guarantees hold across the full training process.

## D FURTHER DATASET DISCUSSION

This section provides further details on the benchmark datasets, AV-GPS and Yahoo Finance.

### D.1 AV-GPS DATASET- FEATURE DETAILS

This section details the AV-GPS dataset (Abrar et al. (2024)). As mentioned in Section 4.1, it is a collection of GPS navigation data collected by the Autonomic Computing Lab GPS-guided Rover (ACL-Rover) at the University of Arizona. Each data point is labelled as **normal (N)** or **attack (A)**, corresponding to the presence of GPS spoofing, which we consider as a sensitive attribute. The dataset comprises four subsets collected under normal and varied spoofing scenarios. Specifically, AV-GPS-Dataset-1 contains 62,042 records (46,287 (N) vs. 15,755 (A)), AV-GPS-Dataset-1-Normal contains 46,287 (N) records, AV-GPS-Dataset-2 totals 6,890 samples (5,184 (N); 1,706 (A)), and AV-GPS-Dataset-3 includes 636 records with 231 (N) and 405 (A) labels. All subsets contain 44 features (– see App D for details on the features). We combined the subsets and utilised 115,855 samples for our experiments. We impute the missing values using the empirical mean and apply min–max normalisation to each feature independently over its entire trace. We then form episodes by segmenting trajectories into fixed-length windows of $T = 100$ steps; each window initialises an episode in which the underlying GPS trace is replayed, but the agent controls the observable stream (via additive perturbations) that an external observer would see. Thus, the agent does not alter the physical trajectory in the dataset, but it does shape the emitted signal on which adversaries operate. Although the dataset is labelled at the window level, we do not train a static classifier on fixed sequences. Instead, the agent issues continuous control actions that update the rover's state and future observations, and the spoofing label is used only for evaluating behavioural leakage via external adversaries. This makes AV–GPS a closed-loop RL control task. Table 2 presents the details of features that are available in the AV-GPS dataset (Abrar et al. (2024)).

### D.2 FINANCIAL TIME–SERIES BENCHMARK DATASET

To evaluate deployment-time behavioural leakage in a realistic financial setting, we construct a benchmark using daily OHLCV (open, high, low, close, volume) data from Yahoo Finance across five highly liquid equities: `AAPL`, `MSFT`, `AMZN`, `GOOG`, and `TSLA`, spanning the period **1 January 2014 to 31 December 2024**. Data collection and cleaning are fully automated using a dedicated script (code will be released upon acceptance), which downloads raw OHLCV records via the `yfinance` API, harmonises column structure, and attaches ticker identifiers. Each record contains prices and volumes for a single trading day. The combined dataset covers more than a decade of price movements across diverse market conditions, providing several thousand usable trajectory windows after segmentation.

We derive a generic, behaviourally meaningful sensitive attribute rather than tying privacy to ticker identity. For each symbol, we first sort records by date and compute daily log-returns on the close price,

$$\text{log\_ret}_t = \log\left(\frac{\text{close}_t}{\text{close}_{t-1}}\right).$$

We then form an $H$-day forward cumulative log-return (with $H=10$ in all experiments),

$$\text{fwd\_log\_ret}_t = \sum_{k=t+1}^{t+H} \text{log\_ret}_k,$$

whenever the full forward window is available. This captures the medium-horizon price movement following day $t$.

For each asset, we construct sliding windows of length $T=100$ trading days. A window starting at day $t$ is represented as

$$X^{(i)} \in \mathbb{R}^{T \times 5},$$

containing the OHLCV sequence from $t$ to $t+T-1$. We associate this window with a binary *sensitive label* based on the forward return *after* the window:

$$y^{(i)} \in \{0, 1\}, \qquad y^{(i)} = \mathbb{I}\big[\text{fwd\_log\_ret}_{t+T-1} > 0\big],$$

Table 2: AV-GPS-Dataset features detail obtained from (Abrar et al. (2024))

| Name | Description |
| --- | --- |
| Roll, Pitch | Rotation around the front-to-back axis of the vehicle is called Roll, and rotation around the side-to-side axis is called Pitch. Measured in degrees, values range from -180° to 180°. |
| Heading | The angular direction of the compass towards which the vehicle's head is pointed, measured in degrees and ranging from 0° to 360°. |
| Yaw, Yaw Rate | Rotation around the vertical axis relative to North, measured in degrees (-180° to 180°). Yaw Rate is the rate of change of Yaw angle with time, measured in degrees/second. |
| Velocity | Actual speed of the vehicle on the ground, is measured in meters/second. |
| Steering Angle | Degree to which the front wheel axle is turned during autonomous navigation, measured in degrees. |
| Relative Altitude | Altitude of the vehicle relative to the initial starting or home location, measured in meters. |
| Altitude AMSL | Altitude above Mean Sea Level (AMSL), with Tucson, Arizona's elevation at approximately 728 meters (2,389 feet). |
| Altitude Tuning, Setpoint | Altitude Tuning converts altitude error to a required climb/descent rate. Setpoints are feed-forward values to altitude controllers, added to outputs and fed back as inputs, primarily for UAVs. |
| X-Track Error | Cross-track error indicating the deviation of the vehicle from its desired track. A gain is applied to converge it to 0. |
| Travelled Distance | Total distance covered by the vehicle after , measured in meters. |
| Run Time | Time record of vehicle's flight or runtime after initialisation, recorded in hours: minutes: seconds format. |
| Distance To Home | Distance to be travelled by the vehicle to reach its home location, measured in meters. |
| Mission Index | Denotes the current mission or waypoint during autonomous mode, with mission planning done beforehand. |
| Heading To Next WP | New heading angle to the next waypoint in the mission, measured in degrees and ranging from 0° to 360°. |
| Heading To Home | Heading angle to the home location relative to the current heading, measured in degrees and ranging from 0° to 360°. |
| Distance To GCS | Distance from the vehicle to the Ground Control Station, measured in meters. |
| Throttle | Percentage of throttle applied by the autopilot to the vehicle's throttle motor, measured in %. |
| Hobbs | Record of "hobbs time," measuring the actual operational time of the vehicle. |
| Clock Time, Clock Date | Current clock time and date. |
| GPS Latitude, Longitude | Current GPS latitude and longitude of the vehicle. |
| GPS MGRS | GPS Military Grid Reference System (MGRS) coordinates for military locating. |
| GPS HDOP, GPS VDOP | Horizontal and vertical Dilution of Precision for GPS signals, indicating positional accuracy. HDOP typically ranges 1-2, VDOP typically 2-4. |
| GPS Course | GPS heading angle or actual direction relative to North, calculated from GPS measures, ranging from 0° to 360°. |
| Satellite Locks, Count | Satellite Lock is the number of satellites the vehicle is connected to, while Satellite Count is the total number available. |
| Longitudinal, Lateral, Vertical Position | 3D coordinates of the vehicle relative to its starting point, measured in meters. |
| Longitudinal, Lateral, Vertical Velocity | 3D coordinate velocities relative to initial velocities, measured in meters/second. |
| Absolute Longitudinal, Lateral Velocity | Absolute values of longitudinal and lateral velocities, measured in meters/second. |
| Temperature | Local temperature of Pixhawk autopilot hardware, measured in Fahrenheit. |
| Longitudinal, Lateral, Vertical Vibration | Vibration levels on the X, Y, and Z axes of the vehicle, measured in Hertz (cycles per second). |
| Data Type | Indicates the type of data: normal data (0) or GPS spoofing attack data (1). |

indicating whether the asset tends to move up ($y^{(i)}=1$) or not ($y^{(i)}=0$) over the subsequent $H$ days beyond the observed window. This label is never revealed to the RL agent and is only used by adversarial models attempting to infer it from the agent's behaviour.

We aggregate over all symbols yields an $(N, 100, 5)$ tensor of windows and an associated label vector $(y^{(i)})_{i=1}^{N}$. Then apply feature-wise min–max scaling across the open, high, low, close, and volume channels. A stratified 80/20 split is performed over the episode labels $y^{(i)}$. This produces disjoint training and test sets where the training split is used for surrogate adversary training and for constructing RL environments where RL agents are always evaluated on held-out trajectories.

# E EXPERIMENTAL ENVIRONMENTAL SETUP ON YAHOO FINANCE DATASET

To evaluate PALADIN beyond autonomous navigation, we construct analogous environments on daily equity time-series drawn from Yahoo Finance using the `yfinance` API Aroussi. As in App D.2, we use OHLCV data [open, high, low, close, volume] for a small universe of highly traded symbols over 2014–2024, sort by (`symbol, date`), and normalise each feature to $[0, 1]$. For each asset, we form sliding windows of length $T=100$ trading days. For the final day of each window, we

compute the $H$-day forward cumulative log-return on the close price (with $H=10$ in all experiments) and assign a binary *sensitive label*

$$y \in \{0,1\}, \qquad y = \mathbb{I}\big[\text{fwd\_log\_ret} > 0\big],$$

indicating whether the asset tends to move up ($y=1$) or not ($y=0$) over the next $H$ days beyond the observed window.

**Environment:**   We define an MDP over these windows. At the start of each episode we sample a window $\tau^{(i)} \in \mathbb{R}^{T \times 5}$ and its label $y^{(i)}$. The state at time $t$ is the normalised OHLCV vector $\mathbf{s}_t \in [0,1]^5$. The agent selects a scalar action $a_t \in [-1,1]$, interpreted as a (long/short) trading position. Let $p_t^{\text{raw}}$ denote the de-normalised close price at time $t$, and let $p_t^{\text{imp}}$ be an internal "impacted price" variable updated by

$$p_{t+1}^{\text{imp}} = p_{t+1}^{\text{raw}} + \text{impact} \cdot a_t,$$

with a small positive impact coefficient. The one-step reward combines P&L and transaction cost:

$$r_t = \big(p_{t+1}^{\text{imp}} - p_t^{\text{imp}}\big) a_t - \text{fee} |a_t|.$$

Thus, while the underlying OHLCV path is a deterministic replay of historical data, the agent's actions affect the impacted price and hence the realised reward trajectory. The environment returns the normalised OHLCV vector $\mathbf{s}_t$ as the observation; the corresponding raw vector $\mathbf{s}_t^{\text{raw}}$ is retained internally for diagnostics.

**Adversarial Leakage Estimation:**   Behavioural privacy leakage is estimated by adversaries trained to classify trajectories into their sensitive labels $y$. As in the AV–GPS setting, we instantiate three architectures:

- **MLP**: a two-layer perceptron with hidden size 256 applied to flattened trajectories;
- **GRU**: a recurrent model with a single GRU layer (hidden size 128), mapping the final hidden state to logits;
- **Transformer**: a Transformer encoder with $L=2$ layers, $H=4$ attention heads, and hidden size 128, with stepwise features linearly embedded, processed by self-attention, mean-pooled, and classified.

All adversaries are trained with cross-entropy loss

$$\mathcal{L}_{\text{priv}} = -\tfrac{1}{M} \sum_{i=1}^{M} \log p_\psi \left( y_{\text{true}}^{(i)} \mid \tau_{1:T}^{(i)} \right),$$

optimised using Adam with learning rate $10^{-3}$. For robustness, we also evaluate agents against mismatched adversaries (e.g. training with an MLP surrogate but evaluating with a Transformer).

**Privacy–Reward Integration and Curriculum:**   We wrap the finance environment with a privacy–reward mechanism. Given the history $\tau_{1:t} = (\mathbf{s}_1, a_1, \ldots, \mathbf{s}_t, a_t)$, the surrogate adversary produces class probabilities $p_w(\cdot \mid \tau_{1:t})$. We define a scalar leakage proxy as the maximum softmax confidence

$$\ell_{\text{priv}}(\tau_{1:t}) = \max_c p_w(c \mid \tau_{1:t}),$$

and form the shaped reward

$$r'_t = r_t - \lambda_t \ell_{\text{priv}}(\tau_{1:t}) - \beta \, \text{MSE}\big(\tilde{\mathbf{s}}_t, \mathbf{s}_t^{\text{raw}}\big), \tag{12}$$

where $\lambda_t$ is a curriculum-driven penalty and $\beta$ is a small fidelity weight. Here $\mathbf{s}_t^{\text{raw}}$ is the normalised OHLCV vector returned by the environment and $\tilde{\mathbf{s}}_t$ is its PALADIN-transformed version used by the policy. Thus, the fidelity term penalises large perturbations but, due to the small $\beta$, contributes only mildly to the overall shaped reward. The penalty schedule $\lambda_t$ follows a staged curriculum (starting from $\lambda_0=0$ and ramping to $\lambda_K$ after early task mastery), so the agent first learns a profitable trading policy before experiencing strong privacy pressure.

**Evaluation:** For each train–test adversary pairing, we train RL agents (baseline, static noise on observations/actions, `adv_no_cur`, `dp_rl`, `dp_nash`, PALADIN) using the shaped reward in equation 12. We then roll out on-policy trajectories from the trained policies and train held-out "true attacker" networks (MLP, GRU, Transformer) on these trajectories. Privacy metrics—adversary confidence (`leak_conf`), negative log-likelihood (`leak_nll`), attack accuracy/F1, and AUROC—are computed post hoc from these held-out attackers, matching a deployment-time threat model in which an external observer trains their own inference model. We also report mean return and standard deviation across seeds, and we conduct $\lambda$-sweeps and ablations (no transform, no curriculum, shallow $f_\phi$, small/weak adversary) to isolate the contribution of each PALADIN component.

# F  FURTHER EVALUATION RESULTS

This section presents additional evaluation details and results for the AV-GPS domain. App F.1 reports a $\lambda$-sweep for PALADIN to characterise the effect of the privacy penalty schedule on both leakage and adversarial performance, followed by App F.2 reporting an ablation study that isolates the contributions of PALADIN's core components. We also present detailed experimental results for the FT domain App F.3. In both cases, we report not only leakage metrics (`leak_conf`, `leak_nll`) but also the attacker's accuracy, F1 score, and AUROC.

## F.1  $\lambda$-SWEEP FOR AVGPS

Table 3 reports the full PALADIN $\lambda$-sweep for all selected train/test adversary pairings. Consistent with the main text, we observe that moderate privacy weights typically yield the best privacy–utility trade-offs, whereas extreme values either collapse utility or offer little additional leakage reduction.

Table 3: $\lambda$-sweep for PALADIN on the AV-GPS dataset (means over five seeds). Higher return and lower leakage are better; attack metrics summarise the strength of the held-out adversary.

| Train/Test Pair | $\lambda$ | Return | $\sigma_{\text{return}}$ | leak_conf | $\sigma_{\text{conf}}$ | leak_nll | $\sigma_{\text{nll}}$ | Att Acc | Att F1 | Att A |
|---|---|---|---|---|---|---|---|---|---|---|
| MLP–MLP | 0.0 | 29.81 | 1.42 | 0.104 | 0.299 | 5.017 | 1.702 | 0.16 | 0.138 | 1.000 |
| | 0.1 | 19.94 | 4.47 | 0.959 | 0.195 | 4.668 | 1.155 | 1.00 | 1.000 | 1.000 |
| | 0.2 | 12.58 | 3.90 | 0.978 | 0.139 | 3.358 | 0.721 | 0.98 | 0.895 | 0.922 |
| | 0.5 | 25.29 | 1.81 | 0.997 | 0.006 | 3.235 | 0.006 | 0.98 | 0.956 | 0.997 |
| | 1.0 | 17.32 | 3.85 | 0.976 | 0.139 | 4.042 | 0.742 | 0.98 | 0.949 | 0.909 |
| | 2.0 | 16.43 | 3.71 | 0.998 | 0.001 | 4.825 | 0.001 | 1.00 | 1.000 | 1.000 |
| MLP–GRU | 0.0 | 23.36 | 3.15 | 0.815 | 0.220 | 0.501 | 0.427 | 0.84 | 0.457 | 1.000 |
| | 0.1 | 35.60 | 5.25 | 0.725 | 0.309 | 0.820 | 0.760 | 0.84 | 0.457 | 1.000 |
| | 0.2 | 25.29 | 2.46 | 0.808 | 0.222 | 0.646 | 0.553 | 0.94 | 0.485 | 0.950 |
| | 0.5 | 40.09 | 5.95 | 0.716 | 0.297 | 0.780 | 0.711 | 0.86 | 0.462 | 1.000 |
| | 1.0 | 27.61 | 5.92 | 0.757 | 0.264 | 0.755 | 0.643 | 0.88 | 0.468 | 0.848 |
| | 2.0 | 15.22 | 8.40 | 0.791 | 0.243 | 0.735 | 0.613 | 0.90 | 0.474 | 0.996 |
| MLP–Transformer | 0.0 | 38.68 | 1.76 | 0.958 | 0.177 | 2.222 | 1.453 | 1.00 | 1.000 | 1.000 |
| | 0.1 | 21.12 | 4.22 | 0.974 | 0.138 | 1.969 | 1.283 | 1.00 | 1.000 | 1.000 |
| | 0.2 | 27.56 | 6.71 | 0.936 | 0.231 | 2.073 | 1.579 | 0.98 | 0.968 | 0.968 |
| | 0.5 | 18.62 | 7.31 | 0.992 | 0.024 | 1.407 | 0.470 | 0.95 | 0.939 | 0.939 |
| | 1.0 | 20.15 | 1.31 | 0.955 | 0.194 | 2.101 | 1.795 | 1.00 | 1.000 | 1.000 |
| | 2.0 | 10.90 | 7.30 | 0.994 | 0.006 | 1.686 | 0.430 | 1.00 | 1.000 | 1.000 |
| Transformer–GRU | 0.0 | 28.47 | 4.04 | 0.988 | 0.008 | 2.578 | 0.009 | 1.00 | 1.000 | 1.000 |
| | 0.1 | 18.16 | 7.57 | 0.986 | 0.004 | 1.518 | 0.004 | 0.94 | 0.867 | 0.857 |
| | 0.2 | 25.43 | 2.35 | 0.979 | 0.003 | 1.635 | 0.003 | 1.00 | 1.000 | 1.000 |
| | 0.5 | 27.76 | 2.70 | 0.947 | 0.190 | 3.779 | 0.812 | 1.00 | 1.000 | 1.000 |
| | 1.0 | 25.76 | 3.20 | 0.991 | 0.003 | 1.612 | 0.003 | 1.00 | 1.000 | 1.000 |
| | 2.0 | 36.96 | 6.66 | 0.924 | 0.226 | 3.993 | 0.856 | 1.00 | 1.000 | 1.000 |
| GRU–Transformer | 0.0 | 25.20 | 4.58 | 0.996 | 0.003 | 1.382 | 0.003 | 1.00 | 1.000 | 1.000 |
| | 0.1 | 21.11 | 2.95 | 0.960 | 0.180 | 3.494 | 0.818 | 1.00 | 1.000 | 1.000 |
| | 0.2 | 38.30 | 1.44 | 0.977 | 0.138 | 3.190 | 0.679 | 0.98 | 0.895 | 0.901 |
| | 0.5 | 25.54 | 3.51 | 0.768 | 0.242 | 5.477 | 0.479 | 0.74 | 0.653 | 0.864 |
| | 1.0 | 14.75 | 3.11 | 0.958 | 0.187 | 3.937 | 0.856 | 0.98 | 0.949 | 0.955 |
| | 2.0 | 26.92 | 2.49 | 0.993 | 0.024 | 3.496 | 0.027 | 0.98 | 0.949 | 1.000 |

## F.2 ABLATION STUDY ON AV-GPS

Table 4 reports the full ablation results for all train/test adversary pairings. As mentioned in Section 5, we observe that `no_curriculum` variants produce highly unstable or negative returns while leaving leakage high; variants without an expressive transform (`no_transform`, `shallow_phi`) achieve moderate utility but remain leaky; and weakening the adversary (`small_adv`, `weak_adv`) largely removes privacy pressure on the policy. These trends are qualitatively consistent across pairings and support the design choices in PALADIN.

Table 4: Ablation study on the AV-GPS Dataset. Att Acc=Att Accuracy, and Att A = Attack AUROC

| Train/Test pair | Method | Return | $\sigma_{\text{return}}$ | leak_conf | $\sigma_{\text{conf}}$ | leak_nll | $\sigma_{\text{nll}}$ | Att Acc | Att F1 | Att A |
|---|---|---|---|---|---|---|---|---|---|---|
| MLP–MLP | no_curriculum | -36.15 | 249.70 | 0.978 | 0.139 | 4.342 | 0.718 | 1.00 | 1.000 | 1.000 |
| | no_transform | 15.15 | 2.08 | 0.977 | 0.139 | 3.738 | 0.800 | 0.98 | 1.000 | 0.994 |
| | shallow_phi | 27.68 | 5.82 | 0.222 | 0.317 | 3.234 | 1.930 | 0.08 | 0.074 | 0.603 |
| | small_adv | 37.26 | 1.81 | 0.996 | 0.002 | 3.210 | 0.001 | 1.00 | 1.000 | 1.000 |
| | weak_adv | 23.38 | 1.05 | 0.998 | 0.000 | 3.250 | 0.000 | 1.00 | 1.000 | 1.000 |
| MLP–GRU | no_curriculum | -764.76 | 1696.78 | 0.768 | 0.284 | 0.541 | 0.478 | 0.67 | 0.457 | 0.716 |
| | no_transform | 18.73 | 4.07 | 0.803 | 0.213 | 0.478 | 0.442 | 0.67 | 0.485 | 0.734 |
| | shallow_phi | 22.39 | 1.34 | 0.784 | 0.252 | 0.529 | 0.419 | 0.67 | 0.479 | 0.746 |
| | small_adv | 22.30 | 5.18 | 0.819 | 0.220 | 0.471 | 0.455 | 0.67 | 0.468 | 0.742 |
| | weak_adv | 25.05 | 2.16 | 0.828 | 0.200 | 0.453 | 0.439 | 0.67 | 0.474 | 0.740 |
| MLP–Transformer | no_curriculum | -1.92 | 102.09 | 0.977 | 0.122 | 1.816 | 1.358 | 1.00 | 1.000 | 1.000 |
| | no_transform | 24.75 | 2.84 | 0.995 | 0.003 | 1.902 | 0.004 | 1.00 | 1.000 | 1.000 |
| | shallow_phi | 14.12 | 6.33 | 0.976 | 0.139 | 3.112 | 0.698 | 0.98 | 0.942 | 0.949 |
| | small_adv | 25.04 | 3.53 | 0.995 | 0.002 | 1.913 | 0.003 | 0.96 | 0.961 | 0.969 |
| | weak_adv | 37.49 | 3.09 | 0.987 | 0.060 | 2.478 | 0.606 | 0.98 | 0.956 | 0.971 |
| Transformer–GRU | no_curriculum | -20.00 | 30.02 | 0.965 | 0.171 | 2.840 | 1.533 | 1.00 | 1.000 | 1.000 |
| | no_transform | -35.89 | 92.44 | 0.977 | 0.139 | 2.940 | 1.153 | 1.00 | 1.000 | 1.000 |
| | shallow_phi | 0.77 | 8.70 | 0.992 | 0.047 | 2.511 | 0.323 | 0.97 | 0.968 | 0.969 |
| | small_adv | -14.23 | 65.24 | 0.994 | 0.005 | 2.646 | 0.056 | 1.00 | 1.000 | 1.000 |
| | weak_adv | 1.52 | 8.62 | 0.972 | 0.143 | 3.069 | 0.765 | 1.00 | 1.000 | 1.000 |
| GRU–Transformer | no_curriculum | -64.48 | 42.84 | 0.977 | 0.139 | 3.153 | 0.778 | 1.00 | 1.000 | 1.000 |
| | no_transform | 22.71 | 8.79 | 0.975 | 0.139 | 2.965 | 0.783 | 1.00 | 1.000 | 1.000 |
| | shallow_phi | 15.10 | 12.71 | 0.995 | 0.005 | 2.114 | 0.010 | 1.00 | 1.000 | 1.000 |
| | small_adv | 14.66 | 4.18 | 0.992 | 0.029 | 2.274 | 0.094 | 1.00 | 1.000 | 1.000 |
| | weak_adv | 39.23 | 2.14 | 0.997 | 0.001 | 1.850 | 0.001 | 1.00 | 1.000 | 1.000 |

## F.3 EVALUATION RESULTS FOR YAHOO FINANCE

Table 5 reports results on the Yahoo Finance trading environment described in App E, across all train–test adversary backbone pairings. The absolute returns are small (of order $10^{-2}$) due to the rewards being computed on normalised OHLCV windows and averaged per step. Therefore, our discussion focuses on *relative* differences in utility and leakage. The **MLP–Transformer** configuration is more adversarially challenging, where returns for all methods remain close to zero, with static action noise and `adv_no_cur` achieving the largest gains ($\approx 0.013$) but only weak leakage suppression. PALADIN slightly trades utility for privacy where it attains a near-zero return, but notably reduces adversary confidence (`leak_conf` drops from $\approx 0.55$ to $\approx 0.48$) and lowers attack F1 (from $\approx 0.74$ to $\approx 0.72$) and increases `leak_nll`; attack accuracy and AUROC also dip slightly (from $\approx 0.59/0.53$ to $\approx 0.57/0.48$). This illustrates a characteristic pattern for strong Transformer attackers where PALADIN can soften the adversary's discriminative performance (lower F1 and confidence), but improvements are not uniform across all leakage metrics. DP-RL marginally reduces F1 further but offers no utility gain and does not exploit the structured privacy penalty used by PALADIN. In the **MLP–GRU** case, the plain RL baseline is slightly loss-making (negative mean return) and behaviourally leaky (attack F1 $\approx 0.81$). Static observation noise attains the highest return in this block but with large variance; it already softens leakage relative to the baseline (lower `leak_conf`, lower attack accuracy and F1, and higher `leak_nll`), so naive noise can be surprisingly competitive here. PALADIN moves into a path with small but positive returns while simultaneously reducing `leak_conf` (from $\approx 0.52$ to $\approx 0.49$) and substantially lowering attack F1 (from $\approx 0.81$ to $\approx 0.70$), and further increasing `leak_nll`; attack accuracy and AUROC drop from $\approx 0.69/0.53$ to $\approx 0.54/0.52$. However, compared to DP baselines, PALADIN achieves better F1 at comparable or higher returns, suggesting a more favourable privacy–utility trade-off when defending

against GRU adversaries. On the other hand, for the **GRU–Transformer** pairing, PALADIN attains the highest average utility in the block (return $\approx 0.041$ versus $\approx 0.005$ for baseline) and modestly reduces attack F1 and accuracy (from $\approx 0.75/0.60$ down to $\approx 0.73/0.58$), but this comes at the cost of slightly lower `leak_nll` and higher `leak_conf`. Hence, privacy gains concentrate in the classification metrics rather than all leakage indicators. This shows that confidence increases (from $\approx 0.49$ to $\approx 0.53$), again highlighting trade-offs between the three leakage metrics. Static noise and `adv_no_cur` either underperform in return or fail to improve F1. DP-RL reduces F1 slightly more than PALADIN but offers almost no improvement in utility. In terms of the **Transformer–GRU** setting, PALADIN again exhibits a strong privacy–utility profile. It achieves the highest return in this block (around $2.1 \times 10^{-2}$, compared to $\approx 1.2 \times 10^{-2}$ for baseline) and clearly reduces attack F1 and accuracy (from $\approx 0.79/0.66$ down to $\approx 0.73/0.57$), while AUROC also drops slightly (from $\approx 0.50$ to $\approx 0.48$). This improvement is accompanied by higher `leak_conf` and lower `leak_nll`, again highlighting trade-offs between the different leakage metrics. `adv_no_cur` attains slightly stronger privacy on the classification metrics (lower F1 and accuracy) but at noticeably lower return, and neither noise-based nor DP-based methods achieve PALADIN's combination of the highest utility with substantially reduced F1 and accuracy relative to the baseline. Finally, in the **MLP–MLP** pairing, the strongest utility in this block actually comes from `adv_no_cur` (return $\approx 2.7 \times 10^{-2}$), with the baseline slightly behind (return $\approx 1.4 \times 10^{-2}$) and PALADIN closer to the privacy-focused end (return $\approx 2 \times 10^{-3}$). The baseline exhibits moderate leakage (`leak_conf` $\approx 0.51$, `leak_nll` $\approx 0.68$, attack accuracy/F1/AUROC $\approx 0.66/0.79/0.52$). PALADIN reduces attack F1 and accuracy relative to both baseline and `adv_no_cur` (F1 from $\approx 0.79/0.75$ down to $\approx 0.71$, accuracy from $\approx 0.66/0.60$ down to $\approx 0.55$), with AUROC slightly higher than baseline, but it does so at the cost of lower return and somewhat higher `leak_conf`. Static noise and `adv_no_cur` therefore act mainly as utility boosters, whereas PALADIN trades some utility for stronger suppression of the classifier-style leakage metrics in this relatively simple backbone pairing, which we primarily treat as a training sanity check.

Overall, Table 5 shows that, under the Yahoo Finance time-series environment, PALADIN delivers utility comparable to or better than strong baselines in the more challenging GRU- and Transformer-based adversary configurations, while consistently lowering attack F1 and accuracy relative to the plain RL baseline. Its impact on `leak_conf`, `leak_nll`, and AUROC is pair-dependent, reflecting the differing inductive biases of MLP, GRU, and Transformer attackers: in some pairings PALADIN improves all leakage metrics, whereas in others it primarily helps on the classifier-based measures (F1/accuracy/AUROC) while trading off confidence or `leak_nll`. In contrast, static noise and DP-based methods provide only modest or inconsistent privacy gains and do not exploit the structure of behavioural trajectories, reinforcing the need for dedicated behaviour-aware shaping.

### F.3.1 Effect of the Privacy Penalty $\lambda$

We further investigate the privacy penalty weight $\lambda$ by sweeping $\lambda \in \{0.0, 0.05, 0.1\}$ for PALADIN across all adversary pairings (Table 6). Because the base environment is relatively well-behaved, the sweeps induce nuanced but informative shifts in the privacy–utility frontier.

In the **MLP–Transformer** case, the Transformer attacker is strong but not dominant. Here, $\lambda = 0.05$ yields the highest mean return in the sweep, but the leakage metrics (`leak_conf`, `leak_nll`, attack accuracy/F1/AUROC) move only slightly relative to $\lambda = 0.0$, indicating that most of the gain at this setting is in utility rather than privacy. A larger penalty $\lambda = 0.1$ lowers adversary confidence and F1 further, but at the cost of reduced return and a higher `leak_nll`. This suggests that in highly expressive adversary regimes, mild penalties help disentangle task-relevant and sensitive structure, whereas overly strong penalties over-regularise the policy without consistently improving privacy. For **MLP–GRU**, the effect of $\lambda$ is more pronounced. Without privacy penalties ($\lambda = 0.0$), PALADIN's return is slightly negative and attacks F1 relatively high. Introducing modest penalties ($\lambda = 0.05$ and $0.1$) turns returns positive and simultaneously reduces F1, with $\lambda = 0.1$ offering both the best utility in the sweep and the lowest `leak_conf`. Although `leak_nll` increases at this setting, the overall trend indicates that a moderate privacy signal is needed to push the agent out of leaky local optima in this backbone pairing. While in the **GRU–Transformer** and **Transformer–GRU** pairings, the sweeps produce subtler changes. For GRU–Transformer, increasing $\lambda$ from $0.0$ to $0.1$ improves utility but does not consistently reduce F1, highlighting that strong Transformer attackers can remain difficult to fool even when privacy pressure is applied. Attack accuracy and AUROC follow the same pattern, fluctuating within a narrow band rather than showing a monotone privacy gain. In the

Table 5: Experimental Results on the Yahoo Finance dataset. Att Acc=Att Accuracy, and Att A = Attack AUROC

| Train/Test Pair | Method | Return | $\sigma_{\text{return}}$ | leak_conf | $\sigma_{\text{conf}}$ | leak_nll | $\sigma_{\text{nll}}$ | Att Acc | Att F1 | Att A |
|---|---|---|---|---|---|---|---|---|---|---|
| MLP-MLP | baseline | 0.014 | 0.019 | 0.514 | 0.096 | 0.683 | 0.195 | 0.655 | 0.792 | 0.515 |
| | static_obs | 0.017 | 0.078 | 0.498 | 0.095 | 0.716 | 0.193 | 0.59 | 0.742 | 0.509 |
| | static_act | 0.016 | 0.042 | 0.547 | 0.077 | 0.614 | 0.152 | 0.625 | 0.769 | 0.564 |
| | adv_no_cur | 0.027 | 0.045 | 0.496 | 0.096 | 0.719 | 0.191 | 0.595 | 0.746 | 0.538 |
| | dp_rl | 0 | 0.001 | 0.545 | 0.079 | 0.619 | 0.163 | 0.545 | 0.706 | 0.573 |
| | dp_nash | 0 | 0 | 0.48 | 0.099 | 0.755 | 0.202 | 0.565 | 0.722 | 0.616 |
| | PALADIN | 0.002 | 0.061 | 0.544 | 0.09 | 0.624 | 0.182 | 0.55 | 0.71 | 0.568 |
| MLP-GRU | baseline | -0.007 | 0.055 | 0.515 | 0.087 | 0.678 | 0.176 | 0.685 | 0.812 | 0.53 |
| | static_obs | 0.04 | 0.139 | 0.449 | 0.063 | 0.81 | 0.131 | 0.55 | 0.706 | 0.501 |
| | static_act | 0.02 | 0.031 | 0.549 | 0.05 | 0.605 | 0.098 | 0.62 | 0.759 | 0.471 |
| | adv_no_cur | 0.008 | 0.018 | 0.529 | 0.064 | 0.644 | 0.127 | 0.64 | 0.774 | 0.544 |
| | dp_rl | -0.002 | 0.013 | 0.529 | 0.074 | 0.647 | 0.145 | 0.6 | 0.747 | 0.561 |
| | dp_nash | 0 | 0.001 | 0.531 | 0.079 | 0.646 | 0.161 | 0.575 | 0.727 | 0.453 |
| | PALADIN | 0.005 | 0.016 | 0.494 | 0.08 | 0.718 | 0.163 | 0.54 | 0.697 | 0.52 |
| MLP-Transformer | baseline | 0.002 | 0.016 | 0.553 | 0.059 | 0.599 | 0.117 | 0.59 | 0.742 | 0.532 |
| | static_obs | -0.002 | 0.013 | 0.515 | 0.073 | 0.675 | 0.147 | 0.6 | 0.75 | 0.489 |
| | static_act | 0.013 | 0.069 | 0.549 | 0.067 | 0.609 | 0.136 | 0.58 | 0.734 | 0.558 |
| | adv_no_cur | 0.013 | 0.021 | 0.506 | 0.079 | 0.694 | 0.158 | 0.56 | 0.718 | 0.533 |
| | dp_rl | 0 | 0.001 | 0.518 | 0.079 | 0.671 | 0.159 | 0.55 | 0.71 | 0.532 |
| | dp_nash | 0 | 0 | 0.536 | 0.071 | 0.632 | 0.142 | 0.61 | 0.758 | 0.571 |
| | PALADIN | -0.001 | 0.005 | 0.475 | 0.074 | 0.756 | 0.148 | 0.565 | 0.722 | 0.475 |
| Transformer-GRU | baseline | 0.012 | 0.052 | 0.515 | 0.087 | 0.679 | 0.176 | 0.655 | 0.792 | 0.498 |
| | static_obs | 0.007 | 0.008 | 0.54 | 0.082 | 0.628 | 0.165 | 0.625 | 0.769 | 0.449 |
| | static_act | 0.001 | 0.009 | 0.503 | 0.09 | 0.703 | 0.182 | 0.565 | 0.722 | 0.518 |
| | adv_no_cur | 0.007 | 0.008 | 0.554 | 0.072 | 0.601 | 0.145 | 0.53 | 0.693 | 0.471 |
| | dp_rl | 0 | 0 | 0.523 | 0.085 | 0.663 | 0.171 | 0.575 | 0.73 | 0.418 |
| | dp_nash | 0 | 0.001 | 0.538 | 0.082 | 0.634 | 0.166 | 0.595 | 0.746 | 0.557 |
| | PALADIN | 0.021 | 0.047 | 0.568 | 0.059 | 0.572 | 0.117 | 0.57 | 0.726 | 0.476 |
| GRU-Transformer | baseline | 0.005 | 0.02 | 0.487 | 0.08 | 0.733 | 0.161 | 0.595 | 0.746 | 0.54 |
| | static_obs | 0.019 | 0.033 | 0.486 | 0.081 | 0.735 | 0.163 | 0.595 | 0.746 | 0.498 |
| | static_act | 0 | 0.024 | 0.561 | 0.039 | 0.58 | 0.073 | 0.545 | 0.702 | 0.47 |
| | adv_no_cur | -0.003 | 0.049 | 0.506 | 0.064 | 0.689 | 0.13 | 0.58 | 0.732 | 0.583 |
| | dp_rl | 0 | 0.003 | 0.502 | 0.075 | 0.7 | 0.151 | 0.565 | 0.72 | 0.53 |
| | dp_nash | 0 | 0.002 | 0.509 | 0.073 | 0.686 | 0.147 | 0.57 | 0.726 | 0.532 |
| | PALADIN | 0.041 | 0.087 | 0.529 | 0.063 | 0.644 | 0.127 | 0.575 | 0.728 | 0.54 |

Transformer–GRU case, small adjustments of $\lambda$ shift returns and leakage metrics within a narrow band; $\lambda = 0.05$ yields the lowest F1, while $\lambda = 0.1$ offers the best return and lowest leak_conf at a slight cost in leak_nll. Finally, for the **MLP–MLP**, the penalty-free variant ($\lambda = 0.0$) achieves moderate return and the lowest attack F1 in the sweep, together with a relatively low AUROC. Increasing $\lambda$ to 0.1 slightly improves mean return (exceeding the baseline utility), whereas $\lambda = 0.05$ underperforms both in return and privacy; in both cases, attack F1 and accuracy increase relative to $\lambda = 0.0$, while leak_nll shifts only modestly.

Across all pairings, the sweeps confirm that *small* privacy penalties ($\lambda \approx 0.05$–0.1) are often sufficient to recover or slightly improve returns while achieving non-trivial changes in leakage metrics, particularly F1. There is no single optimal $\lambda$: different adversary backbones favour different points along the privacy–utility frontier, supporting our use of curriculum-scheduled penalties rather than a fixed, globally tuned value.

### F.3.2 ABLATION STUDY ON YAHOO FINANCE DATASET

We next dissect PALADIN's components via ablations on the Yahoo Finance environment (Table 7), evaluating variants that remove or weaken key design elements: the transformation network (no_transform), the curriculum (no_curriculum), the adversary (small_adv, weak_adv), and the depth of the feature mapping (shallow_phi). For the **MLP–MLP** pairing, removing the curriculum (no_curriculum) leads to negative mean returns only mixed changes in privacy metrics relative to PALADIN, showing that unscheduled penalties destabilise training even in this relatively simple setting. Eliminating the transformation network (no_transform) preserves reasonable returns but leaves trajectories highly identifiable (higher F1, higher attack accuracy, and higher AUROC than PALADIN). Shallow feature mappings (shallow_phi) yield the highest returns among ablations but with the worst attack F1, higher attack accuracy, and elevated leak_conf, indicating that shallow obfuscations primarily smooth trajectories rather than truly hiding the sensitive label. In the **MLP–Transformer** setting, the stronger adversary accentuates these trends.

Table 6: $\lambda$-sweep for PALADIN on the Yahoo Finance Dataset. Att Acc=Att Accuracy, and Att A = Attack AUROC

| Train/Test Pair | $\lambda$ | Return | $\sigma_{\text{return}}$ | leak_conf | $\sigma_{\text{conf}}$ | leak_nll | $\sigma_{\text{nll}}$ | Att Acc | Att F1 | Att A |
|---|---|---|---|---|---|---|---|---|---|---|
| MLP-MLP | 0.0 | 0.012 | 0.015 | 0.538 | 0.1 | 0.639 | 0.204 | 0.55 | 0.71 | 0.562 |
| | 0.05 | 0.002 | 0.008 | 0.494 | 0.111 | 0.731 | 0.229 | 0.625 | 0.769 | 0.579 |
| | 0.1 | 0.021 | 0.032 | 0.497 | 0.102 | 0.719 | 0.2 | 0.63 | 0.772 | 0.598 |
| MLP-GRU | 0.0 | -0.003 | 0.047 | 0.529 | 0.075 | 0.648 | 0.15 | 0.625 | 0.765 | 0.529 |
| | 0.05 | 0.01 | 0.013 | 0.54 | 0.078 | 0.628 | 0.157 | 0.555 | 0.712 | 0.536 |
| | 0.1 | 0.015 | 0.036 | 0.46 | 0.071 | 0.787 | 0.145 | 0.55 | 0.706 | 0.491 |
| MLP-Transformer | 0.0 | 0.007 | 0.075 | 0.514 | 0.08 | 0.678 | 0.162 | 0.6 | 0.75 | 0.525 |
| | 0.05 | 0.027 | 0.065 | 0.533 | 0.077 | 0.641 | 0.156 | 0.62 | 0.765 | 0.547 |
| | 0.1 | 0.01 | 0.023 | 0.483 | 0.079 | 0.741 | 0.159 | 0.58 | 0.734 | 0.454 |
| Transformer-GRU | 0.0 | 0.011 | 0.019 | 0.501 | 0.096 | 0.709 | 0.194 | 0.565 | 0.722 | 0.495 |
| | 0.05 | 0.008 | 0.011 | 0.501 | 0.092 | 0.709 | 0.187 | 0.55 | 0.71 | 0.497 |
| | 0.1 | 0.011 | 0.03 | 0.496 | 0.09 | 0.718 | 0.184 | 0.575 | 0.73 | 0.538 |
| GRU-Transformer | 0.0 | -0.008 | 0.036 | 0.521 | 0.07 | 0.662 | 0.14 | 0.565 | 0.717 | 0.545 |
| | 0.05 | 0.003 | 0.019 | 0.522 | 0.079 | 0.662 | 0.16 | 0.635 | 0.775 | 0.52 |
| | 0.1 | 0.004 | 0.017 | 0.529 | 0.074 | 0.647 | 0.148 | 0.585 | 0.738 | 0.488 |

`No_curriculum` and weakened adversaries (`small_adv`, `weak_adv`) achieve moderate returns but fail to meaningfully reduce F1, and generally maintain high attack accuracy and AUROC, with `leak_nll` moving in ways that do not correspond to a clear privacy gain. `No_transform` achieves good utility but leaves leakage metrics close to or worse than baseline. In contrast, full PALADIN delivers very low adversary confidence and the lowest $F1$ among the ablated variants, together with reduced attack accuracy and AUROC, at modest cost to return, confirming that both the transformation network and the curriculum are active contributors to privacy gains. While for **MLP–GRU**, ablations highlight the joint importance of a strong adversary and a non-trivial feature mapping. `No_curriculum` and `no_transform` both give near-zero or negative returns and leakier behaviours than PALADIN. Variants with weakened or smaller adversaries (`small_adv`, `weak_adv`) can achieve higher returns, but their attack F1 remains only very slightly better than PALADIN's and comes with worse behaviour in other leakage metrics (e.g., higher `leak_conf` or lower `leak_nll`), revealing fragile trade-offs that depend on how strongly the adversary shapes the policy. In the **GRU–Transformer** and **Transformer–GRU** pairings, removing curriculum again degrades performance: `no_curriculum` produces lower or negative returns and does not systematically improve leakage metrics. `No_transform` and weakened adversaries attain moderate returns but leave attack F1 high and do not exploit the full potential of trajectory-space obfuscation. Shallow mappings (`shallow_phi`) occasionally deliver good returns but consistently underperform PALADIN in at least one leakage metric, particularly F1.

**To sum up, the ablations support our design hypothesis that PALADIN's effectiveness on financial time-series arises from the *combination* of (i) curriculum-scheduled privacy penalties, (ii) sufficiently expressive feature transformations, and (iii) strong adversarial supervision.** Removing curriculum leads to unstable or low-utility policies; removing or weakening the transformation network leaves the agent behaviourally leaky, with higher attack F1, accuracy, and often AUROC; and weakening the adversary produces privacy–utility trade-offs that are brittle and backbone-specific. In contrast, the full PALADIN configuration consistently lies on or near the empirical privacy–utility frontier across Yahoo Finance experiments, combining competitive returns with the strongest overall suppression of classifier-based leakage (F1, accuracy, AUROC), whereas static noise and DP-based baselines do not.

PALADIN's success in this financial setting stems from its ability to learn *where* and *when* to obfuscate within OHLCV windows: it discovers which price–volume fluctuations are most predictive of future-movement labels and selectively masks them only after a reward-focused warmup. This targeted approach contrasts sharply with blanket noise or parameter-level DP, which cannot discriminate signal from noise in a dynamic market environment. Moreover, PALADIN avoids the instability of fixed-penalty adversarial shaping by ramping up privacy pressure only after the policy has converged to a strong utility baseline.

However, several limitations remain. First, our Yahoo Finance benchmark uses pre-normalised OHLCV windows and deterministic replay, which simplifies dynamics relative to live trading. Real-

world deployment would require handling non-stationary markets and partial observability. Second, while we show cross-backbone robustness within a single-agent trading setup, extending PALADIN to multi-asset strategies or continuous portfolio rebalancing is non-trivial and may require richer transformation networks and risk-aware objectives.

Table 7: Ablation study on the Yahoo Finance Dataset. Att Acc=Attack Accuracy, and Att A = Attack AUROC

| Train/Test Pair | Method | Return | $\sigma_{\text{return}}$ | leak_conf | $\sigma_{\text{conf}}$ | leak_nll | $\sigma_{\text{nll}}$ | Att Acc | Att F1 | Att A |
|---|---|---|---|---|---|---|---|---|---|---|
| MLP-MLP | no_transform | 0.011 | 0.03 | 0.458 | 0.092 | 0.799 | 0.188 | 0.645 | 0.784 | 0.525 |
| | no_curriculum | -0.021 | 0.029 | 0.529 | 0.11 | 0.66 | 0.222 | 0.55 | 0.71 | 0.514 |
| | small_adv | -0.001 | 0.017 | 0.501 | 0.092 | 0.707 | 0.186 | 0.545 | 0.706 | 0.543 |
| | shallow_phi | 0.048 | 0.101 | 0.561 | 0.08 | 0.589 | 0.16 | 0.68 | 0.81 | 0.513 |
| | weak_adv | 0.004 | 0.019 | 0.533 | 0.094 | 0.646 | 0.186 | 0.56 | 0.718 | 0.565 |
| MLP-GRU | no_transform | -0.018 | 0.044 | 0.481 | 0.077 | 0.744 | 0.156 | 0.605 | 0.751 | 0.508 |
| | no_curriculum | -0.006 | 0.015 | 0.546 | 0.067 | 0.613 | 0.132 | 0.635 | 0.775 | 0.425 |
| | small_adv | 0.037 | 0.05 | 0.558 | 0.052 | 0.588 | 0.097 | 0.545 | 0.696 | 0.574 |
| | shallow_phi | 0.003 | 0.017 | 0.52 | 0.076 | 0.665 | 0.15 | 0.595 | 0.744 | 0.571 |
| | weak_adv | 0.024 | 0.026 | 0.548 | 0.042 | 0.605 | 0.078 | 0.535 | 0.695 | 0.474 |
| MLP-Transformer | no_transform | 0.014 | 0.018 | 0.515 | 0.082 | 0.676 | 0.165 | 0.575 | 0.73 | 0.555 |
| | no_curriculum | 0 | 0.004 | 0.519 | 0.075 | 0.667 | 0.15 | 0.6 | 0.75 | 0.563 |
| | small_adv | 0.01 | 0.021 | 0.548 | 0.066 | 0.609 | 0.133 | 0.62 | 0.765 | 0.543 |
| | shallow_phi | 0.029 | 0.041 | 0.451 | 0.066 | 0.807 | 0.134 | 0.61 | 0.758 | 0.489 |
| | weak_adv | 0.015 | 0.028 | 0.501 | 0.08 | 0.703 | 0.162 | 0.605 | 0.754 | 0.533 |
| Transformer-GRU | no_transform | -0.033 | 0.128 | 0.525 | 0.078 | 0.656 | 0.159 | 0.655 | 0.792 | 0.483 |
| | no_curriculum | -0.02 | 0.03 | 0.524 | 0.088 | 0.662 | 0.178 | 0.625 | 0.769 | 0.525 |
| | small_adv | -0.012 | 0.038 | 0.493 | 0.086 | 0.722 | 0.175 | 0.595 | 0.746 | 0.534 |
| | shallow_phi | 0.006 | 0.022 | 0.516 | 0.087 | 0.676 | 0.175 | 0.595 | 0.746 | 0.512 |
| | weak_adv | 0.004 | 0.033 | 0.514 | 0.087 | 0.681 | 0.176 | 0.61 | 0.758 | 0.476 |
| GRU-Transformer | no_transform | -0.005 | 0.031 | 0.525 | 0.068 | 0.654 | 0.136 | 0.6 | 0.747 | 0.499 |
| | no_curriculum | 0.006 | 0.024 | 0.517 | 0.083 | 0.673 | 0.167 | 0.605 | 0.754 | 0.582 |
| | small_adv | 0.015 | 0.035 | 0.504 | 0.077 | 0.698 | 0.155 | 0.54 | 0.695 | 0.619 |
| | shallow_phi | 0.022 | 0.095 | 0.548 | 0.064 | 0.609 | 0.128 | 0.63 | 0.77 | 0.556 |
| | weak_adv | 0.007 | 0.029 | 0.502 | 0.083 | 0.702 | 0.167 | 0.57 | 0.724 | 0.51 |

# G  BROADER IMPACTS

By embedding privacy directly into the RL loop, PALADIN offers practitioners a principled way to mitigate behavioural data leakage in sensitive applications such as autonomous vehicles, personalised healthcare and financial trading. This proactive defence can enhance user trust, comply with data-protection regulations (e.g. GDPR), and reduce the risk of surveillance or identity inference from smart-agent behaviours.

On the other hand, adversarial privacy shaping could be misused to conceal malicious or unsafe agent behaviours (e.g., evading auditing in critical systems) or to obfuscate policy actions in adversarial settings such as cybersecurity. Therefore, care must be taken to balance privacy with accountability and transparency, and to ensure that privacy-preserving agents remain subject to appropriate oversight and testing before deployment.

