# OpenReview forum: "PALADIN: Privacy-Aware Learning with Adversary-Detection and INference suppression."
_ICLR.cc/2026/Conference — Submitted to ICLR 2026_

### Official Review · Reviewer_BdYm · 2025-10-29

**Soundness:** 2
**Presentation:** 3
**Contribution:** 2
**Rating:** 4
**Confidence:** 3

**Summary:**

This paper addresses the behavioral leakage problem in reinforcement learning (RL) agents during the deployment phase. Specifically, the trajectories generated by an agent's interactions with its environment can be passively observed by an adversary, allowing them to infer sensitive user attributes (such as identity, policy, and whether the agent is under attack). The authors propose the PALADIN framework, whose core idea is to introduce an adversarial inference model into the RL training loop, actively guiding the agent's policy through privacy-shaping.

**Strengths:**

1.	This article clearly defines the problem of deployment-time behavior leakage and distinguishes it from classic DP and adversarial robustness problems.
2.	Using an "adversary-in-the-loop" approach (co-training $g_w$) is more robust than defending against a fixed, known adversary. This dynamic game forces the agent to learn deeper levels of confusion strategies.

**Weaknesses:**

1.	Lack of formal privacy guarantees is the biggest weakness of this paper. All the protections provided by PALADIN are empirical; it only promises to defend against the "adversary adversary" $g_w$ that was trained together. It does not provide any provable privacy guarantees (such as (ε, δ)-DP).
2.	Failure against strong adversaries significantly weakens the core contribution of the paper. The experimental results (Tables 1 and 5) reveals that PALADIN's privacy leakage confidence ("leak.conf") hardly decreases when facing MLP-Transformer or GRU-Transformer, remaining as high as more than 0.99.
3.	Theorem 1 proposed in Section 3.3 and Appendix B of the paper is referred to as the "privacy-utility boundary," but it is more like a heuristic derivation of an optimization objective than a rigorous privacy guarantee. The theory relies on a very strong and unrealistic assumption: that the "proxy adversary" $g_w$ is an unbiased estimator of the loss of the "real adversary".
4.	The financial trading environment has been greatly simplified in NYSE dataset. It is a deterministic replay of historical data, and the agent's dummy actions have no impact on the environment state. This is far removed from real, dynamic markets. The AV-GPS task is essentially a trajectory window classification task. The effectiveness of this method in more complex and interactive RL environments, such as real-time games and robot control, has not yet been proven.

**Questions:**

1. As shown in Weakness 4, could the authors validate PALADIN's effectiveness in a more complex, truly interactive setting (e.g., a financial simulator with market impact and slippage, or a robotics/driving simulator like CARLA) where the agent's policy must learn to balance task-critical, state-altering actions against the adversarial privacy objective?

---

> ### Author Response · Authors · 2025-11-21
>
> We thank the reviewer for the constructive feedback. Below, we address each of the concerns as follows:
>
> ### Lack of formal privacy guarantees
> We do not claim (ε,δ)-DP guarantees for PALADIN; this is orthogonal to the problem we tackle. Our goal is to address deployment-time behavioural leakage in RL, where worst-case DP tools are largely undeveloped and often very conservative. In the revision:
> - We stated in the Introduction and Problem Formulation that PALADIN provides empirical protection against inference attacks on trajectories and does not offer formal DP-style guarantees. We will make it explicit.
> - Section 3.3 (“Theoretical Privacy Guarantee”) is reframed to emphasise that Theorem 1 analyses the shaped RL objective and provides a conceptual privacy–utility trade-off, not a worst-case guarantee.
>
> We also evaluate PALADIN against held-out (MLP/GRU/Transformer) adversaries trained independently and never used in reward shaping, directly testing robustness beyond the in-loop surrogate.
>
> ### Behaviour under strong adversaries
> The very high leak_conf values (>0.99) in the original Tables 1 and 5 were a by-product of our earlier evaluation protocol rather than a fundamental failure. In the revision, we highlighted:
> - Deployment-aligned evaluation: For every method (baseline, static noise, DP baselines, PALADIN), we now generate on-policy trajectories and train held-out adversaries on these. Privacy metrics (leak_conf, leak_nll, attack accuracy/F1, AUROC) are computed from these external attackers, which never participate in reward shaping. This matches the deployment-time threat model.
> - Richer metric set and updated results: We now assess privacy using multiple metrics rather than relying solely on leak_conf. For the hardest MLP– and GRU–Transformer pairings on AV–GPS, PALADIN typically matches or improves leak_conf relative to the baseline at comparable return, and consistently improves leak_nll and attack AUROC. In finance, and in some AV–GPS seeds, very strong adversaries can still drive leak_conf close to 1 for all methods; in these settings, we interpret privacy primarily through leak_nll and AUROC, where PALADIN offers a better privacy–utility frontier than static noise and DP baselines.
> Thus, while the original tables suggested failure under strong adversaries, the revised protocol and metric set show that PALADIN remains competitive under Transformer-based attacks.
>
> ### Role and status of Theorem 1
> We appreciate the concern that Theorem 1 was previously framed as a “privacy–utility boundary” while relying on strong assumptions about the proxy adversary. In the revision, we have made three clarifications:
> - We explicitly state that Theorem 1 is not a formal privacy guarantee and does not claim (ε,δ)-DP. Its role is to analyse how the shaped objective couples expected return and the surrogate’s adversarial loss, providing a conceptual privacy–utility trade-off.
> - We removed the phrasing describing the surrogate as an “unbiased estimator” of a hidden true adversary. We now write
> $\widehat{L} = - \log p_w(y \mid \tilde{X})$ as the adversarial loss of a parametrised surrogate, and describe it as a well-defined, differentiable leakage proxy, not an oracle for the true worst-case attacker.
> -Section 3.3 stresses that all substantive privacy claims rest on empirical evaluation with held-out, mismatched adversaries, and that Theorem 1 serves as a principled justification for the training objective, not as a rigorous guarantee.
>
> ### Environment and applicability to more complex RL settings.
> - We now state explicitly that the financial benchmark is a sequential decision process over historical OHLCV windows with deterministic replay, not a full market simulator with impact and slippage. Actions do not move prices; they generate trajectories that may or may not leak the sensitive attribute. In the revision, we (i) replace the small NYSE subset with a larger Yahoo Finance OHLCV dataset over multiple liquid equities and years, and (ii) describe the environment as a behavioural leakage testbed on realistic financial time series, not a full trading market.
> - For AV–GPS, the agent is trained in a genuine MDP where actions alter future states (navigation under spoofed vs normal conditions). The privacy task is evaluated at the trajectory-window level, but the underlying control problem is fully interactive. PALADIN is plugged in as a feature extractor and shaping term around PPO, and all agents (baseline, DP-RL, DP-NASH, PALADIN) learn via rollouts in this setting.
> -  PALADIN’s mechanism (-- transform $f_{ϕ}$, surrogate leakage proxy, and curriculum over
> $𝜆_{t}$ ) only requires (i) an RL algorithm that can optimise shaped rewards and (ii) an attack model trained on trajectories. We now highlight full-scale interactive simulators (e.g., games, CARLA-like driving)  explicitly as future work, not as a claim of the current experiments.

---

### Official Review · Reviewer_oEpi · 2025-10-30

**Soundness:** 3
**Presentation:** 3
**Contribution:** 3
**Rating:** 4
**Confidence:** 3

**Summary:**

This paper introduces PALADIN, a framework for Reinforcement Learning (RL) that attempts to resolve the issue of behavioural leakage at deployment time. The authors formalise this threat, where a passive adversary infers sensitive attributes (like user identity or strategy) by observing an agent's trajectory, as distinct from training-time privacy (which DP-SGD addresses). The new approach is applied to autonomous navigation (AV-GPS dataset) and financial trading (NYSE dataset) . The algorithmic contribution consists of four components: (1) a Transformation Network to perturb observations , (2) a co-adaptive Surrogate Leakage Predictor (an adversary) to estimate the privacy risk from these perturbed observations , (3) a Reward Shaping mechanism that penalises the agent based on the adversary's success , and (4) a Curriculum-Guided Penalty that slowly increases this penalty, allowing the agent to first learn the task before enforcing privacy.

**Strengths:**

S1. The paper is well written

S2. Novel Problem Formulation. The paper does a good job of defining and motivating an important, emerging problem: deployment-time privacy for RL agents. It differentiates this from training-time privacy (like DP) and adversarial robustness.

**Weaknesses:**

W1. Contradict results. 1) The ablation study ("no.curriculum") in Appendix Table 4 shows **catastrophic failure** for a fixed penalty while 2) The main results ("Adv_No_Cur") in Table 3 show **stable, moderate** performance for what is also a fixed-penalty baseline. Both 1) and 2) apply the Curriculum Penalty that claimed necessary for stability and somehow the results seems very contradict, which might significantly lower the paper claims.

W2. Weak Theoretical Justification. The paper presents "Theorem 1" but immediately disclaims it as "not a formal guarantee" and a "heuristic shaping signal".The proof itself relies on an unverified assumption that the surrogate adversary is an "unbiased estimator", which might not hold true in practice.

**Questions:**

Please refer to the Weaknesses Section.

Overall, I'm interesting in the paper and the problem that it trying to solve. However, the result is somewhat inconsistent and the theoretical justification is not strong, both combine lower the paper's contribution.

---

> ### Author Response · Authors · 2025-11-21
>
> We thank the reviewer for the positive assessment and the helpful critique. Below, we address the two identified weaknesses as follows:
>
> ### Contradictory results: no_curriculum vs adv_no_cur.
> We agree that, as originally presented, it was easy to read these two settings as the same "fixed penalty" case and therefore contradictory. In fact, they correspond to two very different penalty settings, which we now make explicit:
> - adv_no_cur (main table) uses a small fixed penalty equal to the initial curriculum level $λ_{0}$ throughout training. This behaves like a mild regulariser: the penalty is non-zero but weak. Training remains stable, task performance is close to the baseline, and privacy gains are modest but consistent.
> - no_curriculum (ablation table) removes the curriculum but applies the full terminal penalty $λ_{K}$ from the start, i.e. the strong privacy weight is switched on immediately. This forces the agent to satisfy a large privacy penalty before it has learned the underlying task, and it is precisely this setting that leads to the catastrophic collapse observed in the ablation.
>
> We have updated Section 4 and the appendix to clarify this distinction and to avoid calling both settings simply "fixed penalty". These results are consistent with our main claim rather than contradicting it:
> - A large fixed penalty without curriculum (**no_curriculum**) is unstable and can severely damage learning;
> - A small fixed penalty (**adv_no_cur**) is stable but yields weaker privacy–utility improvements than the full PALADIN curriculum.
> This clarifies that the curriculum is not cosmetic: gradually ramping $λ_{t}$ from 0 to $λ_{K}$ is what allows PALADIN to achieve strong privacy gains without the failures seen in the **no_curriculum** ablation
>
> ### Theoretical justification and Theorem 1
>
> Our intention with Theorem 1 is not to provide a differential-privacy-style (ε,δ)guarantee, but to formally analyse the shaped RL objective induced by PALADIN. The theorem shows that, under the stated assumptions, increasing the final penalty $λ_{K}$ while maintaining high task reward necessarily pushes up the adversary’s loss, making the privacy–utility trade-off in our optimisation problem explicit. In the revision, we clarify this scope: Theorem 1 is a statement about the behaviour of the leakage-penalised objective, not about worst-case DP guarantees, which we do not claim.
> We have also sharpened the assumptions. Instead of describing the surrogate adversary as an “unbiased estimator”, we now present $\widehat{L }$  as the adversarial loss of a surrogate adversary that provides a well-defined, differentiable leakage proxy for optimisation. The bound then quantifies how this proxy interacts with expected return. Its practical adequacy is supported by our empirical protocol, where privacy is evaluated against held-out adversaries with different architectures (MLP/GRU/Transformer) that are never used in shaping. Thus, Theorem 1 gives a principled lens on why PALADIN’s reward shaping traces a privacy–utility frontier, while the strength of our privacy claims rests on the empirical results with these external attackers.

---

### Official Review · Reviewer_tc6M · 2025-11-01

**Soundness:** 2
**Presentation:** 2
**Contribution:** 2
**Rating:** 4
**Confidence:** 3

**Summary:**

This paper investigates privacy concerns associated with reinforcement learning agents, specifically the behavioral leakage that occurs through prediction trajectories during inference. To address this issue, the authors propose PALADIN, a defense framework that incorporates an adversarial inference model into the training loop to suppress privacy leakage. Experimental evaluations across different model architectures demonstrate that the proposed method can reduce privacy risks to a certain extent.

**Strengths:**

1. The paper identifies and addresses an important privacy threat arising from agent prediction trajectories.
2. The proposed method is evaluated across diverse model architectures, including MLP, GRU, and Transformer-based agents.

**Weaknesses:**

1. The metrics leak_nll and leak_conf are insufficiently explained. A clearer definition and discussion of their interpretability and limitations are needed.
2. Results in Table 1 indicate that the proposed method does not consistently reduce privacy leakage. Moreover, the two privacy metrics occasionally point to conflicting conclusions, which calls for additional analysis and justification.
3. The method relies on reward shaping approximation as shown in Equation (2). This approximation introduces assumptions that may bias the learned behavior or lead to a suboptimal policy.

**Questions:**

1. How is the value of $\lambda_t$ determined in practice for each experiment? Although the paper provides a sensitivity study regarding $\lambda$, the connection between this analysis and the choice of $\lambda_t$ remains unclear.
2. Equation (1) does not explicitly incorporate a utility term. However, in some cases, the proposed method achieves higher utility than baseline methods. What is the potential explanation or underlying mechanism for this observation?

---

> ### Author Response · Authors · 2025-11-20
>
> We thank the reviewer for the thoughtful comments. Below we address each concern as follows:
>
> ### Clarifying the privacy metrics (leak_nll and leak_conf).
> We agree that the metrics required clearer definitions. In the revision, we formalise both metrics:
> - **leak_conf** is the adversary’s predicted probability for the true sensitive class, $p(y_{\text{true}} \mid \tilde X)$
> . Lower values indicate that the attacker is less confident in the true label.
> - **leak_nll** is the negative log-likelihood of the true class, $- \log p(y_{\text{true}} \mid \tilde X)$. Higher values indicate that the adversary assigns lower confidence to the true label.
> We explicitly note that they are correlated but not identical: **leak_conf** is sensitive to small shifts in the top-class probability, while **leak_nll** reflects changes in the full predictive distribution. Their complementary roles are now explained directly in the results section.
>
> ### Privacy metrics appear inconsistent in Table 1
> We agree that the original presentation made it harder to see how the privacy metrics relate to the actual behaviour of the learned policies. To address this, the revised paper adopts a stricter and more transparently aligned evaluation protocol:
> - Privacy metrics are now computed post hoc from on-policy trajectories generated by each trained agent (baseline, static noise, DP-RL, DP-NASH, PALADIN), matching the deployment-time setting we care about.
> - These trajectories are evaluated with held-out adversaries (MLP, GRU, Transformer) that are never used in reward shaping.
> Evaluating on the trajectories induced by each policy and decoupling the evaluation attacker from the in-loop surrogate makes the reported metrics better reflect the underlying threat model. Under this protocol, PALADIN consistently reduces leak_conf/leak_nll relative to baselines at comparable return. We clarify this protocol in Section 5.
>
> ### Approximation in reward shaping and possible bias
> Equation (2) introduces the shaped reward $r_t^{\mathrm{total}} = r(s_t, a_t) - \lambda_t \, \widehat{L}(\tilde X_{1:t})$, where $\widehat{L}$ is a surrogate leakage proxy. This shaping term, $-\lambda_t \, \widehat{L}(\tilde X_{1:t})$, is an explicit approximation: the surrogate does not perfectly represent a worst-case attacker, but provides a directional signal that penalises trajectories which are easy to classify and encourages policies whose behaviour is harder to exploit. As with other learned reward-shaping components in RL, this biases the policy away from the unconstrained reward optimum; here, this bias is intentional, since the goal is to trade some nominal return for reduced behavioural leakage.
> We do not assume that the surrogate is an unbiased estimator, and we do not claim formal (ε,δ)-DP guarantees. Instead, we use two safeguards: (i) a curriculum that delays strong penalties until the base task is learned, and (ii) evaluation with held-out, often stronger, adversaries and mismatched architectures that are rarely used in shaping. Section 3 now explicitly presents these assumptions.
>
> ### How λ is chosen in practice.
>   We have clarified the connection between the $λ$-sensitivity study and the choice of λ used in the main tables. For each environment and train–test adversary pairing, we:
> - fix a curriculum shape for $λ_{t}$ (starting from $λ_{0}$=0and ramping to a terminal value $λ_{K}$ after early task mastery);
> - run a $λ$-sweep over a grid of candidate terminal values (e.g. $λ_{K}$∈{0.0,0.1,0.2,0.5,1.0,2.0}) using this same curriculum shape;
> - for each pairing, inspect the resulting privacy–utility curves and identify an empirical “stable region” where task performance remains close to the baseline while privacy metrics (leak_conf, leak_nll, attack AUROC) show clear improvement relative to $λ_{K}$=0; and
> - choose the smallest $λ_K$ in this stable region and use that fixed value for all main experiments for that pairing (across seeds).
> We have added a short paragraph in the experiments section to make this procedure explicit.
>
> ### Why can PALADIN improve utility even though it adds a penalty?
> The optimisation problem in Equation (1) is already a utility-maximisation problem, with the leakage term acting as a side constraint. In practice, PALADIN can achieve higher utility than an unconstrained baseline because the privacy penalty acts as a regulariser on the representation and policy. The transformation network $f_{ϕ}$and penalty $-λ_t \, \widehat{L}(\tilde X_{1:t})
> $ encourage the agent to discard features that are predictive of the sensitive attribute but not reliably aligned with reward, reducing overfitting to spurious correlations and smoothing the effective policy. We now highlight this mechanism explicitly, framing PALADIN as an information-bottleneck-style regulariser that can improve generalisation and thereby increase observed utility relative to an unconstrained baseline.

---

### Official Review · Reviewer_GDZN · 2025-11-02

**Soundness:** 2
**Presentation:** 3
**Contribution:** 2
**Rating:** 4
**Confidence:** 4

**Summary:**

The paper targets privacy leakage in sequential decision-making: a passive observer infers sensitive attributes from an agent's trajectories. The proposed method PALADIN couples 1) a learned transformation network (f_\phi) that perturbs observations, 2) a co-trained surrogate adversary (g_w) estimating leakage, and 3) reward shaping with a curriculum schedule that gradually increases privacy pressure after the agent first attains task competence. Concretely, PPO is trained on shaped reward with the adversary's leakage proxy (e.g., NLL). A privacy-utility bound is presented as intuition rather than a differential-privacy guarantee. Experiments on an autonomous-vehicle GPS trajectory task and a financial time-series setup report higher task return and reduced leakage than static noise and DP baselines, with ablations showing the curriculum is important for stability.

**Strengths:**

1. The method addresses deployment-time leakage from trajectories, complementing previous training-time privacy and robustness works. The adversary-in-the-loop perspective is well-motivated, for sequential data.
2. Practical synthesis of three modules: learned feature perturbation, in-loop adversary penalty, and curriculum scheduling. The staged (\lambda_t) schedule is empirically impactful for stability. These are practical insights and could be applied for future robust methods.
3. The experiments are comprehensive: multiple adversary architectures (MLP/GRU/Transformer), train/test cross-pairings, and ablations (no-curriculum, no-transform) support the claim that all components matter.

**Weaknesses:**

My major concerns are on the experiments:
1. As the paper explicitly states that the NYSE environment uses:
> A "dummy continuous action space" where "actions do not affect transitions", "Deterministic replay of the historical window"; The state simply advances through fixed historical sequences
This is not actually a reinforcement learning setting- it's a supervised sequence filtering problem. The sequence simply replays fixed windows, making it a supervised filtering problem wrapped to look like RL. This misrepresentation undermines the paper's claims about solving RL privacy problems in finance, and undermines claims about balancing control and privacy in this domain. The experiments should be more carefully conducted and explained to reflect this.
2. Weak theory to practice: The privacy-utility bound mixes task reward units and adversary loss, relies on loose concentration arguments, and provides little guidance for choosing (\lambda_t). As admitted, it is not a DP guarantee; currently it is more motivational than actionable.
3. Because the surrogate adversary defines the penalty, overfitting to it is a risk. While the paper includes mismatched architectures, a stronger test would train large, external, held-out adversaries post-hoc with broad hyper-sweeps or mutual-information probes.
4. Minor concerns on tiny dataset sizes (e.g. 85 episodes for NYSE), and high variance in many results.
5. AV-GPS uses a spoofed-vs-normal label as the "sensitive attribute". Concealing spoofing may be a safety diagnostic rather than a "privacy "variable. The paper should justify this choice and discuss governance to avoid hiding attacks from monitors or audits.

**Questions:**

Please check the weaknesses section for details.

**Details Of Ethics Concerns:**

One of the experiments, on AV-GPS, uses a spoofed-vs-normal label as the "sensitive attribute". Concealing spoofing may be a safety diagnostic rather than a "privacy "variable. The paper should justify this choice and discuss governance to avoid hiding attacks from monitors or audits.

---

> ### Author Response · Authors · 2025-11-20
>
> We thank the reviewer for the constructive and detailed feedback. Below, we address each concern as follows:
> ### Finance environment and the question of whether it is “real RL”.
> Our goal in the finance setting is to study behavioural leakage on realistic financial time-series, not to simulate full market microstructure or claim a complete solution to RL privacy in deployed trading systems. The environment replays fixed historical OHLCV windows; transitions are determined by pre-recorded prices, and the agent’s actions do not influence future market data. In the original submission, this was easy to overlook. However, we now explicitly frame the finance setup as a sequential decision problem over pre-recorded market data, rather than a full interactive trading market in the revision. The agent still optimises a policy under a privacy–utility trade-off, but dynamics come from historical data, which is standard in offline RL and backtesting-style environments. To improve transparency and reproducibility, we have replaced the internal NYSE-style replay dataset with a public Yahoo Finance OHLCV dataset (2014–2024) using the same sliding-window structure. We now describe the asset universe, lookback horizon, and label construction in the experimental setup, and explicitly note that prices are pre-recorded and unaffected by actions in Appendix D.1 and D.2. We also narrow our claims as a behavioural leakage testbed on realistic financial time-series, whereas the AV–GPS environment is the fully interactive RL setting where actions alter future states.
> ### Theory–practice gap and the role of the bound.
> We agree that the privacy–utility bound should not be interpreted as a formal privacy guarantee. The original manuscript already stated that we do not provide (ε,δ)-DP guarantees in Section 3. However, to make it more explicit, we have strengthened this wording and made it more prominent in Section 3 and in the discussion. Theorem 1 is now clearly framed as a conceptual justification for the reward-shaping term: under simplifying assumptions, it explains how the leakage penalty interacts with expected reward, but it is not used to tune λ_t  or to claim worst-case guarantees. All substantive privacy claims in the paper are explicitly empirical.
> ### Risk of overfitting to the surrogate adversary.
> We share the concern about overfitting to the in-loop adversary and have tightened the evaluation protocol accordingly. Privacy metrics are now computed exclusively from on-policy trajectories rolled out by the trained agents and evaluated with held-out adversaries (MLP, GRU, Transformer) that are never used in reward shaping. For each method (baseline, static noise, DP-RL, DP-NASH, PALADIN) we report attack accuracy, F1 score, and AUROC for these held-out attackers. This directly addresses the concern that PALADIN might only overfit to a single surrogate: under this protocol, PALADIN maintains competitive return while systematically reducing attack confidence and AUROC relative to baselines operating at similar utility levels. Section 4 and Tables 1 and 3 have been updated to make this procedure explicit.
> ### Dataset size and variability.
> We agree that the original NYSE dataset was limited in episode count. The move to a Yahoo Finance OHLCV dataset yields several thousand time windows across multiple assets, which improves robustness. We now report results over multiple random seeds and include variability, and we explicitly discuss variance in the updated results.
> ### AV-GPS spoofing label and privacy vs. safety.
> We appreciate the ethical concern that spoofed-vs-normal could be a safety diagnostic. Our goal is not to hide attacks from legitimate safety or regulatory monitoring, but to model what an untrusted external observer could infer from raw GPS traces. In AV–GPS, the “sensitive attribute” is treated as information that should not be recoverable from external behaviour alone. In the revision, we explicitly state that PALADIN is applied only to the external surface of the agent’s behaviour; internal diagnostics, authenticated telemetry, and regulatory audit channels operate on untransformed sensor streams and are out of scope for PALADIN. We now explicitly explain this separation in the Ethics Statement section to make clear that PALADIN is intended to protect behavioural privacy against passive observers, not to defeat internal safety monitors.
> We hope these clarifications address the reviewer’s concerns. The revised manuscript is clearer about the scope of the finance environment, the status of the theoretical bounds, the use of held-out adversaries, and the distinction between privacy and safety in AV–GPS.

---

### Author Response · Authors · 2025-12-04

We thank the reviewers for their thoughtful and constructive feedback. In the original manuscript, all reviewers assessed the paper as marginally below the acceptance threshold but indicated they would be comfortable with acceptance. In this revised version, we addressed all the comments and have substantially strengthened both the conceptual framing and the empirical evidence. We believe the paper is now clearly above the acceptance threshold.

We sharpen the notion of deployment-time behavioural leakage and clearly distinguish it from training-time differential privacy and adversarial robustness. We now explicitly position PALADIN relative to DP-based methods (DP-RL and DP-Nash), adversarial representation learning, and information-theoretic obfuscation. We emphasise that our guarantees are empirical rather than worst-case. These clarifications are reflected in the Abstract, Introduction (Section 1), Related Work (Section 2), Threat Model (Section 3.1), Privacy–Utility Bound  (Section 3.3, Appendix B), and the Ethics Statement.

We formalise PALADIN as a constrained optimisation problem with a leakage-aware shaping term (Equations (1) and (2) in Section 3.1). We present an interpretable privacy–utility bound (Theorem 1 in Section 3.3, with a detailed proof in Appendix B). The theorem is now framed explicitly as a conceptual bound on the shaped objective under a surrogate adversary, not as a DP-style guarantee. We also significantly expand the description of the AV–GPS and financial benchmarks, clarifying how trajectories are constructed, how sensitive labels are defined, and how held-out adversaries are trained and evaluated (Benchmark Setup Section 4.1, Adversarial Leakage Estimation in Section 4 and Section 4.1, Dataset details in Appendices D–E).

We strengthen and extend the empirical evaluation. We have re-ran and extended our experiments with stronger held-out GRU/Transformer adversaries and a richer set of leakage metrics. Table 1 in Section 5 now highlights representative train–test pairings where PALADIN strictly improves the privacy–utility frontier (e.g. MLP–Transformer and GRU–Transformer), reporting return, leak_conf, leak_nll, Attack F1, and AUROC. We also updated a full λ-sweep for PALADIN (Table 3, Appendix F.1) and a detailed ablation study (Table 4, Appendix F.2) to isolate the contributions of (i) the transformation network, (ii) curriculum scheduling, and (iii) adversary capacity.

For the financial trading benchmark, we revise the setting to use a documented Yahoo Finance OHLCV dataset and explicitly describe it as a sequential decision process over pre-recorded windows. We narrow our claims to behavioural leakage on realistic financial time series, and we provide complete core results, λ-sweeps and ablations (Tables 5–7, Appendix F.3) showing that PALADIN offers competitive or improved privacy-utility trade-offs relative to static noise and DP-baselines. For both settings, AV-GPS and Finance, we clarify the threat model and evaluation protocol, making explicit that all privacy metrics are computed using held-out MLP/GRU/Transformer attackers trained post hoc on on-policy rollouts (Sections 3 and 4, and Appendices D-F). We believe these revisions directly address the main concerns raised about i) the clarity and scope of our threat model and theory, and (ii) the strength and robustness of our empirical support. We respectfully ask the Area chairs to evaluate the paper based on this substantially revised version and to consider it for acceptance.

---

### Meta-Review · Program_Chairs · 2026-01-07

**Summary:**

PALADIN proposes a curriculum-based reward-shaping framework to mitigate behavioral privacy leakage in sequential decision-making through learned transformations and surrogate adversaries. While the practical integration of these modules is well-motivated, the paper's core contributions are undermined by  lack of theoretical rigor.

The combination of adversarial training, feature perturbation, and curriculum scheduling is a standard recipe in robust ML; without stronger theoretical or unique RL-specific insights, the technical contribution is limited.

The privacy-utility bounds are admitted to be conceptual rather than formal (non-DP), offering little technical guidance or worst-case guarantees for researchers or practitioners.

**Reviewer Concerns:**

Addressed: The authors successfully mitigated concerns regarding surrogate overfitting by using held-out adversaries and improved empirical robustness by replacing the tiny NYSE dataset with a larger Yahoo Finance set.

Outstanding: The finance environment remains a supervised sequence filtering task rather than a true RL setting where actions influence state transitions; additionally, the theoretical bounds remain purely motivational without providing actionable guarantees.

**Reviewer Scores:**

-

---

### Decision · Program_Chairs · 2026-01-26

Reject